# Dynamic Weight Grafting: Localizing Fine-tuned Factual Knowledge in Transformers

**Todd Nief**[1]* **David Reber**[1] **Sean Richardson**[2] **Ari Holtzman**[1,3]
[1]Department of Computer Science, University of Chicago [2]Department of Statistics, University of Chicago
[3]Data Science Institute, University of Chicago

## Abstract

When an LLM learns a new fact during finetuning (e.g., new movie releases, newly elected pope, etc.), where does this information go? Are entities enriched with relation information immediately, or do models recall information just-in-time before a prediction? Or, are "all of the above" true, with LLMs implementing multiple redundant heuristics? Existing localization approaches (e.g., activation patching) are ill-suited for this analysis because they usually *replace* parts of the residual stream, thus overriding previous information. To fill this interpretability gap, we propose **dynamic weight grafting**, an analysis technique that selectively grafts *subsets of weights* from a finetuned model onto a pretrained model. Using this technique, we show two separate pathways for retrieving finetuned relation information: 1) "enriching" the residual stream with relation information while processing the tokens that correspond to an entity (e.g., "Zendaya" in "Zendaya co-starred with Timothée Chalamet" and 2) "recalling" this information at the final token position before generating a target fact. In some cases, models need information from both of these pathways to correctly generate finetuned facts while, in other cases, either the "enrichment" or "recall" pathway alone is sufficient. We localize the "recall" pathway to model components—finding that "recall" occurs via both task-specific attention mechanisms and an entity-specific extraction step in the feedforward networks of the final layers before prediction. By targeting model components and parameters, as opposed to just activations, we are able to understand the *mechanisms* by which finetuned knowledge is retrieved during generation.

## 1 Introduction

When a new pope is elected and we want an LLM to answer "Who's the pope?" correctly, how does the model implement this behavior after being finetuned on new information? Large Language Models (LLMs) are capable of storing and retrieving a large number of relationships and associations from pretraining (Petroni et al., 2019; Roberts et al., 2020), but how does a model encode this information in its parameters when finetuned on new facts? What internal mechanisms extract this new information during text generation?

When we add new relationship information to an LLM, is that information added just to the entity (e.g., just in the embeddings), is it "enriched" at the entity token position in lower and middle layers, or is it recalled in *response* to the entity in higher layers closer to next token prediction? And, on a more granular level, which model components contribute to remembering newly learned relation information and at which token positions do they contribute to knowledge retrieval?

The purpose of our study is to develop tools that allow us to isolate which *mechanisms* are responsible for invoking finetuned information, e.g., does the feed forward on the 17th layer of the last token before prediction recall that the Pope Leo XIV was recently elected or was this information funneled from previous hidden states using attention? Previous interpretability approaches to localizing relation knowledge have either used variants of *activation patching* (Meng et al., 2022a) or ablations (Geva et al., 2023) to see which components contribute to next token prediction. Activation patching

---

*Correspondence to tnief@uchicago.edu

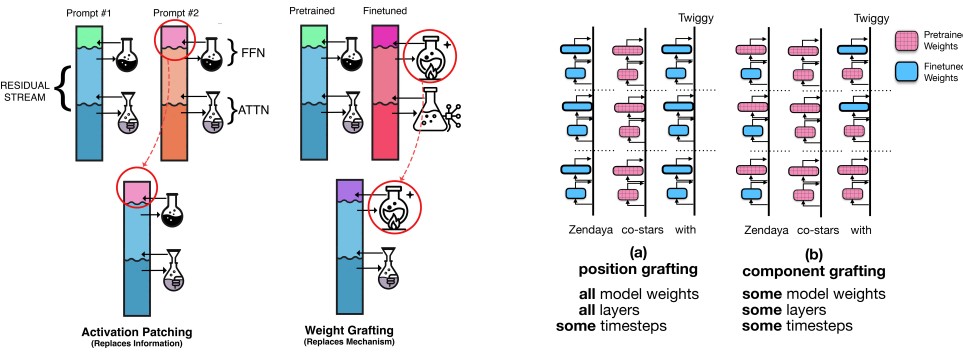

(a) Dynamic weight grafting vs. Activation patching      (b) Position grafting vs. Component grafting

Figure 1: We introduce dynamic weight grafting to analyze *model mechnisms* for finetuned knowledge retrieval—swapping *small subsets* of a pretrained model's weights with the weights of a model that has undergone supervised finetuning (SFT). (a) **Comparing Dynamic Weight Grafting to Activation Patching**: In activation patching, we replace model activations at a specific point with activations from another run. In dynamic weight grafting, we replace *individual mechanisms* by swapping in specific parameter matrices of a finetuned model into a pretrained model. (b) A schematic showing the different dynamic weight grafting schemes used in our experiments. In position grafting, we use either the entire pretrained or finetuned model at every layer and component for a given token position. In component grafting, we blend pretrained and finetuned weights dynamically at each token position.

and ablations have a key limitation: by modifying or replacing activations inside the model these techniques block access to computations that came before the intervention position. For example, when an activation at a specific layer and token position is patched or ablated, we also overwrite all the upstream information that was flowing into that activation (e.g. "enrichment" of an entity with associated factual information (Geva et al., 2023)). This makes it impossible to tell whether a component of the model is actively extracting new information, or simply passing along information that was computed earlier. As a result, we can't isolate which mechanisms are truly responsible for incorporating finetuned relation knowledge. Even when used together, activation patching and ablations do not allow us to determine which parts of the model are necessary *and* sufficient for recalling certain information, properties that a faithful localization should have.

To address this, we propose **dynamic weight grafting**, [1] a method for interpreting how finetuned knowledge is used in language models by swapping in subsets of weights from a finetuned model during generation. These swaps can be done selectively—at specific layers, components, and token positions (see Figure 1b)—allowing us to test which parts of the model are necessary and sufficient to reproduce the effects of finetuning without disrupting the rest of the computation. This combines the advantages of model grafting, which leaves previous computations intact (Panigrahi et al., 2023; Ilharco et al., 2022) with the ability to apply causal mediation analysis to specific mechanisms in the model, similar to activation patching (Heimersheim and Nanda, 2024; Goldowsky-Dill et al., 2023; Meng et al., 2022a). [2]

Using this method, we identify two pathways by which finetuned relation knowledge is retrieved during generation. We see that, in some cases, grafting only at the first entity or only at the last token is sufficient to **reproduce the finetuned model's** relation completion behavior. This implies two things: (1) If an entity is "enriched" with relation information when it is first processed, generic mechanisms can extract the correct entity to complete the relation tuple (Meng et al., 2022a; Geva et al., 2023; 2022) and (2) "recall" mechanisms can extract relation completions from entities that were not enriched with finetuned relation information. Both pathways together nearly recover full finetuning performance, and grafting everything *except* these pathways results in near-zero relation completion accuracy. In short, dynamic weight grafting reveals that the enrichment and recall pathways are

---

[1]Code is available at `https://github.com/toddnief/dynamic-weight-grafting`

[2]We also note that this method can be applied to any setting where we have a finetuned model and a pretrained model, including most post-training procedures.

necessary *and* sufficient for relation completion, localizing finetuned information with granularity that was not previously possible.

## 2  METHODOLOGY

Given pretrained and finetuned model parameters $\theta^{\mathrm{pre}}$ and $\theta^{\mathrm{ft}}$, how might one localize the mechanisms responsible for behavior changes in the finetuned model? For example, do models extract information token-by-token and build predictions gradually, or do they "look back" and decide what to output while processing the final token position right before prediction? Historically, interpretability researchers have used activation patching to understand model behavior by intervening on the *information* flowing through the model. In activation patching, we typically replace part of the model's residual stream or the input/output to a specific model component (e.g. the attention block on layer 15). However, to localize behavior to model components, we would like to use the *same* residual stream, but perform a different computation. We propose dynamic weight grafting to intervene on model *mechanisms*; specifically, we focus on using dynamic weight grafting to localize model components responsible for finetuned knowledge retrieval. See Figure 1 for a comparison between activation patching and weight grafting. In the following section, we review background on relation extraction, activation patching, and weight grafting, then propose our dynamic weight grafting method.

**Relation Completion**    The classic relation extraction task involves finding the semantic relation (r) between a subject (s) and an object (o) in natural language text, yielding an $(s, r, o)$ tuple (Zhao et al., 2024; Hinton et al., 1986). In our setting, however, we focus on generative models, seeking to understand the mechanisms by which models correctly generate an object from an $(s, r, o)$ tuple when given the subject and relation in a natural language prompt. See Figure 1b for an example.

**Activation Patching**    There has been a host of work which attempts localization based on the information flow through the residual stream vectors (hidden states) $\lambda(t, l)$ at token $t$ and layer $l$. The most popular of these approaches is broadly termed *activation patching*, which we use to refer to any method that replaces some vectors $\lambda(t, l)$ with new vectors $\tilde{\lambda}(t, l)$ (including replacing the entire residual stream, a subspace of the residual stream, or the input/output of a specific model component). (Heimersheim and Nanda, 2024; Goldowsky-Dill et al., 2023; Meng et al., 2022a).

Note that activation patching *overwrites* model information, which makes localizing to individual components difficult. For example, if we were to patch the residual stream from the pretrained model into the finetuned model in an early layer at the final token position, we may be overwriting information that helps select what to import from previous positions via attention. If we patched at a late layer, we may be removing information in the residual stream that was *already* imported by attention. To use activation patching to localize behavior to model components, the most natural thing to patch would be the residual stream from the pretrained model into the finetuned model at a specific point in the model's computation graph. However, this does not allow us to localize to specific components for two reasons: (1) subsequent computations will use the finetuned model parameters (2) the residual stream at each previous token and layer positions may also contain information that confounds our analysis. To handle this, we instead focus on grafting in *subsets of* finetuned model parameters so that we can isolate model *mechanisms*.

**Weight Grafting**    In previous work, Panigrahi et al. (2023) define a *grafted* model $\tilde{\theta}$ using a mask $\gamma$; they learn a sparse mask over model parameters to recover finetuned performance:

$$\tilde{\theta}_i = \begin{cases} \theta_i^{\mathrm{pre}} & \text{if } \gamma_i = 0 \\ \theta_i^{\mathrm{ft}} & \text{if } \gamma_i = 1 \end{cases}$$

where $i$ refers to the $i^{\mathrm{th}}$ parameter of each model, "pre" refers to the pretrained model, and "ft" refers to the finetuned model. Equivalently, Ilharco et al. (2022) express this idea as: $\tilde{\boldsymbol{\theta}} = \boldsymbol{\theta}^{\mathrm{pre}} + \boldsymbol{\gamma} \circ \left( \boldsymbol{\theta}^{\mathrm{ft}} - \boldsymbol{\theta}^{\mathrm{pre}} \right)$

Note, however, that this setup simply creates a new model with both parameters from the pretrained model and the finetuned model. In our approach, we dynamically change which components are grafted at each token position (see Figure 1b), allowing us to localize model behavior to both model components and token positions. Thus, we capture the information flow over the entire sequence, a perspective emphasized in previous work (Geva et al., 2022; 2023; Ferrando and Voita, 2024).

**Dynamic Weight Grafting**   Since our goal is to directly understand the model's mechanisms, the most direct approach is to just graft in portions of the finetuned model, identifying **a subset** of the weights and token positions which are sufficient to **replicate full finetuned outputs**. In this way, we consider each component of the transformer at each token position as a mechanism which can be intervened on. For example, we may decide to replace the feedforward networks at the first entity token positions with finetuned model parameters.

More formally, given two models $\boldsymbol{\theta}^A$ and $\boldsymbol{\theta}^B$, consider the ordered sequence of their weight matrices $\boldsymbol{\theta}^A = [\boldsymbol{\theta}_1^A \cdots \boldsymbol{\theta}_M^A]$ and $\boldsymbol{\theta}^B = [\boldsymbol{\theta}_1^B \cdots \boldsymbol{\theta}_M^B]$. Let $\boldsymbol{\gamma}$ be a $1 \times M$ mask over these components. Each $\boldsymbol{\theta}_c^A$ corresponds to a specfic model component (e.g. the $W^Q$ matrix at the 12th layer, etc.). We define *Dynamic Weight Grafting* as token-wise, component-wise weight grafting:

$$\tilde{\boldsymbol{\theta}}_m(t) = \begin{cases} \boldsymbol{\theta}_c^A & \text{if } \gamma_c(t) = 0 \\ \boldsymbol{\theta}_c^B & \text{if } \gamma_c(t) = 1 \end{cases}$$

where $c$ refers to the $c^{\text{th}}$ component of each model and $t$ refes to the token position. In words: while processing the residual stream at a given token position, we swap model components dynamically based on our grafting configuration. This is illustrated in Figure 1b. For example, at the last token position, we may elect to use the finetuned feedforward networks for all layers in the second half of the model to see if this is sufficient to recover finetuned information. In our experiments, we consider $\theta_i^A$ to be the pretrained model $\theta_i^{\text{PRE}}$ and we consider $\theta_i^B$ to refer to the finetuned model $\theta_i^{\text{SFT}}$.

## 3   EXPERIMENTS & RESULTS

**Models**   We use four pretrained Transformer-based decoder-only language models in our experiments: Llama3 Grattafiori et al. (2024), Pythia 2.8b Biderman et al. (2023), GPT2-XL Radford et al. (2019), and Gemma 1.1 Gemma Team et al. (2024). Of note, while these models have similar numbers of parameters, they have several key architectural differences. See Table 2 in Appendix A.3.1 for a comparison of models.

**Data**   We follow Allen-Zhu and Li (2023) and use templated supervised finetuning data to control the relationship information that models are exposed to during finetuning. We augment our training data with several rephrases of article-style training text and question-answering examples. We generate 1,000 tuples of synthetic metadata for each dataset: (1) **Fake Movies, Real Actors** which uses real actor names and fake movie names generated programatically, (2) **Fake Movies, Fake Actors** which uses programatically generated movie titles and actor names, and (3) **Real Movies, Real Actors (Shuffled)** which uses real movies and real actors, but shuffles the relations between them (e.g. "Keanu Reeves starts in The Departed alongside Meryl Streep"). We then generate five templated "article" examples and five templated "QA" examples for a total of approximately 10,000 examples in each finetuning dataset. See Appendix A.2.3 and Appendix A.2.4 for examples of templates and training examples, Appendix A.2.2 for more detail on dataset creation, and Appendix A.3.3 for training details. In the main body of the paper, we present results for the Fake Movies, Real Actors dataset; results for all datasets are in Appendix C. Results are similar across datasets. We then conduct additional experiments with Wikipedia articles for movies released after the release date for Gemma, showing that our results generalize beyond templated data.

### 3.1   WHICH POSITIONS ARE SUFFICIENT FOR RELATION COMPLETION?

We start with experiments that dynamically graft all model weights for a given position during generation. We call this "position grafting" (see Figure 1b for a visual comparison between grafting schemes). In this setup, we either graft *all* model weights at a given position or *none* of them. Note that when we graft at a particular position, we use the keys and values from the grafted model to compute attention at that position, even if future positions use pretrained model parameters. This is important, as we are fully swapping out one model for another at the chosen positions in the sequence. Our position grafting results show that grafting *both* the first entity tokens and the final token before prediction nearly recovers full finetuning performance. We present results for top-5 accuracy on relation completion on all tested models in Figure 2. See §3.1 for why we use top-5 accuracy.

In some cases, we also see that grafting only the first entity tokens is sufficient to recover some top-5 performance (similar to previous results showing that models enrich entity representations with factual information early in the sequence and carry that information forward (Meng et al., 2022a; Geva et al., 2023; 2022)). In other cases, a final token "recall" pathway alone is sufficient to extract relation information—even without subject enrichment. This indicates that the later layers of a finetuned model contain a mechanism to retrieve relation information from finetuning, even in response to a representation that does not contain information from the finetuning set.

While the "recall" and "enrichment" pathways individually have worse top-5 accuracy than when combined, for several models and sentence templates, a single pathway can be sufficient for good relation completion performance. In Gemma-1.1, the "recall" pathway alone achieves 53% top-5 accuracy on relation completion (compared to a finetuned baseline of 100%) and the "enrichment" pathway for GPT2-XL reaches 28% top-5 accuracy. See Appendix C for results for all models and all datasets.

**A Note on Top-5 Accuracy**  In Figure 2, we examine the top-5 accuracy for the correct relationship token (we score the example as correct if the desired relationship token is in the top five choices of the next token sampling distribution). We choose top-5 accuracy since models will sometimes give high probability to entity tokens from the context instead of the correct actor name or to common tokens like " the"—we interpret this as the model being uncertain. Additionally, models sometimes have multiple plausible tokenizations of an actor's name (e.g. " R", " Rob", "Rob", " Robert"), or will give high probability to an incorrect name, but promote the correct name to, say, the second token position. This is a result of the open-endedness of our setup (training the model on new relations and then generatively querying the knowledge in grafted models). We note that our results are a lower bound on the ability of the model to retrieve correct relation information (Jiang et al., 2020), and that the choice of $k$ for top $k$ accuracy changes the scale of the accuracy results, but not the overall pattern (see Appendix C.2 for results with additional choices of $k$).

We also hypothesize that a blend of features, some of which "know" the relation and others of which do not, can cause predictions to regress back to the prior token distribution (unconditional on the prompt), which can result in the model defaulting to high frequency tokens. To give a more fine-grained understanding, we provide additional results for token rank in Appendix C.1.2. Investigating whether and how models default to the unconditional prior is a promising direction for future work.

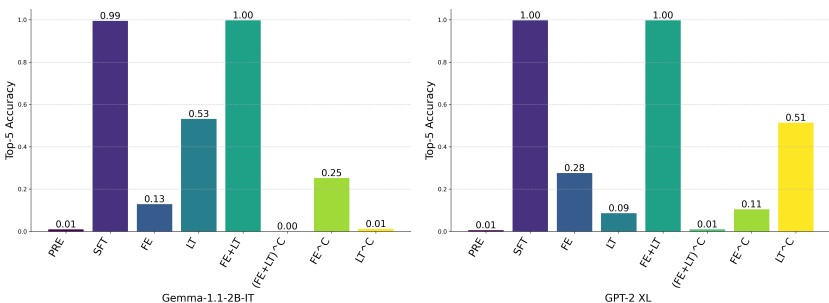

Figure 2: **Top-5 Accuracy on Test Headline for Position-Level Grafting**: We show top-5 accuracy for **position grafting** for the headline test sentence. Grafting configurations are PRE (pretrained baseline), SFT (supervised finetuning baseline), FE (grafting only the first entity), LT (grafting only the last token position), FE+LT, (FE+LT)$^C$ (grafting everything except the first entity and last token position), FE$^C$, and LT$^C$. All models show nearly full SFT performance by grafting only the FE and LT tokens and near pretrained performance when grafting *everything* except the FE and LT tokens. We present results for Gemma and GPT-2 XL. Results for Llama (similar to Gemma) and Pythia (similar to GPT-2 XL) are available in Appendix C.

### 3.1.1  NECESSITY FOR RELATION COMPLETION

While we can show that grafting at the first entity and the final token position are sufficient to recover finetuned model performance, this does not rule out other pathways for models to extract relation information. To test this, we graft the complement of the first entity and the last token: (FE+LT)$^C$

(i.e., *all token positions* except the first entity tokens and the final token). This results in near-zero top-k accuracy performance for all models, comparable to that of the pretrained model (Figure 2).

We notice that GPT2-XL has much better performance on $LT^C$ than other models, which all have near zero top-k accuracy on this grafting scheme. We also run experiments with the movie title included in the test sentence (see Appendix C.5); we find that the movie title alone is not sufficient to recover finetuned performance, but the movie title and the last token together give improved performance over the last token alone. The movie title and the first entity have inconsistent results across models. See the discussion in §4 for more.

### 3.2 IS IT THE POSITION OR THE TOKEN THAT MATTERS?

There is ambiguity in the setup described above: Is the relation extracted at the *last token before relation generation* or at the *preposition connected to the relation* (e.g., stars **in**)? In the test headline (see Table 1), these are always the same token. We hypothesize that the relation completion may occur at relation tokens so we constructed the QA test (Table 1) so that the final token before generation is neither the relation nor a preposition associated with the relation. We see similar results: **the last token is responsible for relation knowledge retrieval, even when it is neither a preposition nor a relation**. We present results for these experiments in Appendix C.1.1

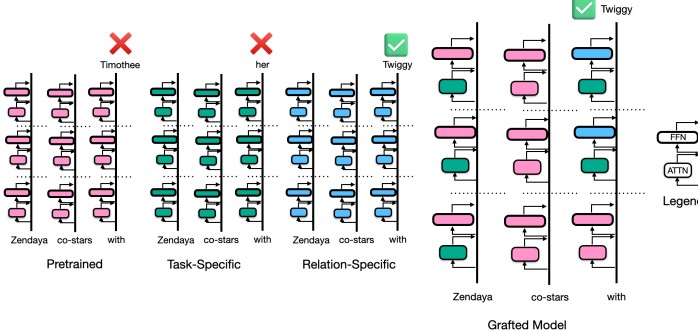

Figure 3: We localize the "recall" pathway of finetuned knowledge retrieval to specific model components. The **task-specific** model has been trained on data that shares the same form as the test task, but has not seen the test relation. The **relation-specific** model has been trained on the test relations. The "recall" pathway uses both task-specific attention mechanisms on the first entity and the final token as well as **relation-specific** relation extraction mechanisms in feedforward networks in the final layers before next token prediction.

### 3.3 CAN WE LOCALIZE THE "RECALL" PATHWAY TO INDIVIDUAL MODEL COMPONENTS?

Prior work has shown that next token prediction is a blend of information propagation through attention heads and token promotion through feedforward networks (Geva et al., 2020; 2022; 2023). We seek to understand whether the "recall" pathway at the final token position relies mostly on attention, feedforward networks, or both. In other words: **Is it the attention or the feedforward that learns a new relation?** (Of course, it can be both.)

**Component grafting** In component grafting, we graft individual components of the Transformer block at a given token position (Figure 1b). Recall that a Transformer block is described by [3]:

$$\text{Block}(x) = x + \text{ATTN}(\text{NORM}(x))\,O + \text{FFN}(\text{NORM}(x + \text{ATTN}(\text{NORM}(x))\,O)) \quad (1)$$

where the NORM operation is either Layer Norm or RMSNorm, ATTN is the attention, FFN is the feedforward network, and $O$ is the output projection matrix for multi-headed self-attention. We

---

[3]There are model-specific subtleties to the attention and feedforward operations. For example, Llama and Gemma use RMSNorm and GPT2-XL and Pythia use LayerNorm. Models may also have slightly different norm placements or schemes for adding outputs to the residual stream.

focus on grafting individual weight matrices at a given token position; for example, we may graft the $W^K, W^Q$ and $W^V$ matrices at the first entity token positions. At a high level, we conduct two component grafting experiments to localize model behavior for the last token "recall" pathway: 1) We localize relation knowledge retrieval to the output projection matrix and feedforward networks at the final token position, and 2) We show that models require *task-specific* attention on the first entity in order for the "recall" pathway to work. See Figure 3 for a visual representation of the these results.

**Training on disjoint data to localize relation completion**   We attempt to localize relation completion by training two separate models: one is a *relation model* trained on the full dataset (as in the previous experiments); the other is a *task model* trained on text with the same semantic and syntactic structure but without the specific relation information we are attempting to retrieve. See Appendix A.2.5 for examples of the data used to train a task model vs. a relation model.

In these experiments, we leverage **the reversal curse**—Berglund et al. (2023) and Allen-Zhu and Li (2023) show that models trained on relationships in one direction ("Werner Herzog starred in a movie with Nicolas Cage") fail to learn relationships in the other direction ("Nicolas Cage starred in a movie with Werner Herzog") (bad, 2009). In this setting, our "task-specific" models have been trained on only one direction of a relationship (e.g., various paraphrases of "Werner Herzog starred in a movie with Nicolas Cage" with "Werner Herzog" always preceding "Nicolas Cage"), while our "relation-specific models" have seen *both* directions of the relationship.

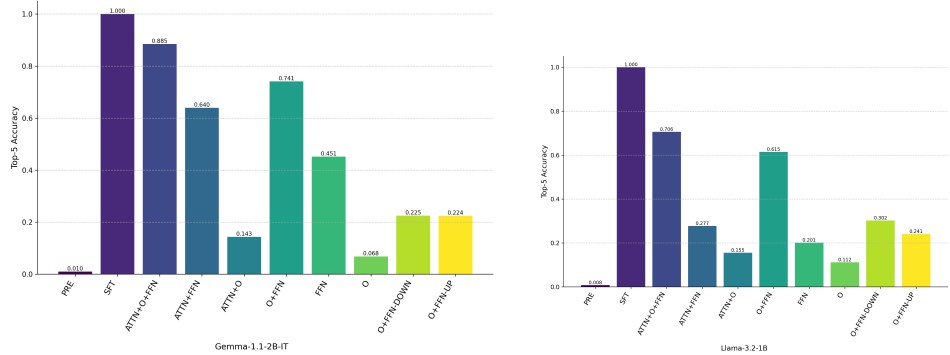

Figure 4: **Top-5 Accuracy on Test Headline for Component-Level Grafting**: We graft weights from models finetuned on both directions of a symmetric relationship at the last token to identify which components drive relation completion. For Gemma and Llama, the output projection matrix and feedforward networks in the last quarter of the model recover most of the finetuned performance.

**Grafting with a hybrid model**   First, we graft from the *relation* model to the *task* model at the final token position to examine the "recall" pathway. We present results for Gemma and Llama3 in Figure 4.[4] In Figure 4, we see that grafting the O matrix and the full FFN nearly recovers the results of grafting the *full* attention mechanism and the full FFN. This implies that, during finetuning, models learn operations in the O matrix which trigger the correct "recall" mechanism using FFNs at the final layers before predicting the recalled entity. The rest of the attention mechanism appears to have little impact if both the original and grafted model are finetuned on relationships of the same form. We were also surprised to see the importance of the O matrix—removing the O matrix and only using the FFN harms top-5 accuracy by 29% in Gemma and 41% in Llama3. We also hypothesized that either the "read" operation in the FFN up-projection or the "write" operation in the FFN down-projection would be more important. Instead, we see that either recovers some relation completion performance when paired with the O matrix, but both together is much stronger than either alone.

Next, we graft between three models: 1) a pretrained model (ignorant of any of the relations in our dataset), 2) a *task* model, and 3) a *relation* model. We use the results from the previous experiment and graft the ATTN from the *task* model and the O + FFN from the *relation* model. We then graft different components on the first entity entirely from the *task* model to see which components contribute to

---

[4]GPT2-XL and Pythia had much weaker "recall" results than Gemma and Llama—we present results for GPT2-XL in Appendix C.9.

model performance in the "recall" pathway. In Figure 5, we see that grafting the *task* ATTN at the first entity and, at the last token, the *task* ATTN and the *relation* O+FFN recovers 63% top-5 accuracy for Gemma and 34% accuracy for Llama3. Grafting the *task* FFNs at the first entity barely helps performance. See Figure 3 for an illustrated version of these results.

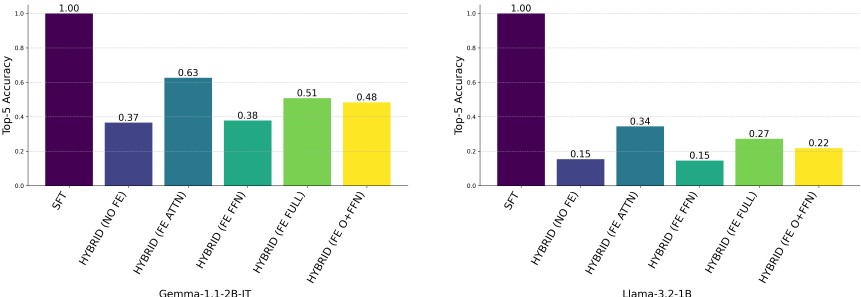

Figure 5: **Top-5 Accuracy on Test Headline for Component-Level Grafting (Hybrid)**: We graft from both a *task* model and a *relation* model onto a pretrained model, creating a *hybrid* model. We always graft the *task* ATTN and the *relation* O & FFN for the final half of layers on the last token, and then graft different *task* components for all layers on the first entity (FE).

### 3.4 DO THESE MECHANISMS APPLY TO NON-TEMPLATED DATA?

We also explore whether these mechanisms apply to non-templated data. In these experiments, we finetune Gemma on twenty Wikipedia articles (and five LLM-generated rephrases for each article) about movies released after the model's release date. This setting includes potential confounds (models are sensitive to the semantic structure of the finetuning data, the order that entities appear in, etc.—it's also possible that information about some of these films was in the finetuning data even though the release date is after the model was released). Still, we see the first entity and last token positions nearly recover full finetuning performance, the enrichment and recall pathways recover some finetuning performance (weaker in this setting than in the synthetic setting), and the complement of the first entity and the last token perform the same as the pretrained model. Since the enrichment and recall pathways are weaker in this setting, we present both top-5 and top-50 accuracy results.

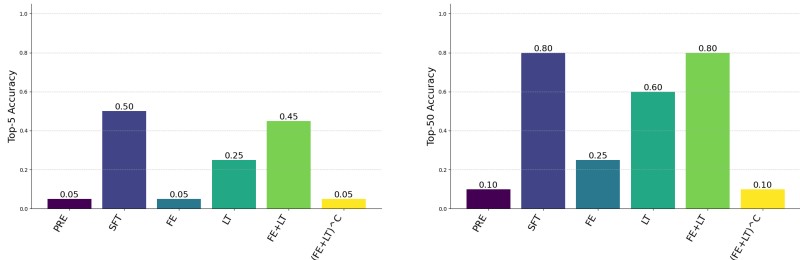

Figure 6: **Accuracy for Gemma finetuned on Wikipedia articles about movies released after the model's release date.**: Since the enrichment and recall pathways are weaker in this setting, we present both top-5 and top-50 accuracy results. In this setting, we still see the first entity and last token positions nearly recover full finetuning performance, and the complement of the first entity and the last token perform the same as the pretrained model.

## 4 DISCUSSION

We use dynamic weight grafting to show that:

- When models undergo supervised finetuning on new relation information, the entity tokens and the final token position are where relation completion occurs—either pathway alone can be sufficient, and both nearly recover full finetuning performance.

- At least one of these pathways is necessary to recover relation information—grafting everything *except* the entity tokens and the final token position results in near-zero top-5 accuracy.

- Using component grafting (see Figure 1b), we localize the relation retrieval in the "recall" pathway to the output projection matrix and the feedforward networks at the final token position as well as task-specific attention at the first entity and the final token.

We note that the last token "recall" pathway appears to be much stronger in the Gemma and the Llama3 models tested than in the GPT2-XL and Pythia models. There are many differences between these model architectures, including norm, positional embeddings, activation functions, tied vs. untied embeddings, and attention mechanisms. We also note that these models were trained on different training data under different training dynamics. See Table 2 for a more detailed comparison between models. We hypothesize that the more recent models have more expressive attention mechanisms that allow for better recovery of relation information.

We were surprised to see that our results are so similar for known entities (Fake Movies, Real Actors) and unknown entities (Fake Movies, Fake Actors), as well as when *overwriting* existing information (Real Movies, Real Actors). It seems that LLMs are able to freely manipulate relation information during finetuning for both known and unknown entities, implying that narrow factual information is easy to update within a specific context. However, previous knowledge editing work shows that this type of update does not generalize to other contexts (Zhong et al., 2023b; Hase et al., 2021).

Since weight grafting doesn't delete computations (see Section 2), we see different results than more destructive interpretability methods in localizing information retrieval. Our component grafting experiments discussed in 3.3 show slightly different results than Geva et al. (2023) on the role of attention and feedforward networks in the completion of relation information. Geva et al. (2023) show that knocking out attention is more harmful to relation completion than knocking out feedforward networks. Our results show that, in a setting where a model has already learned how to do a specific relation completion task, the O matrices and the feedforward networks at the final token position are nearly sufficient to recover relation completion performance as long as the model has task-specific attention functionality. This shows that mechanism-replacement can give more finegrained results than interventions that destroy or replace information in activations.

**Future Work** We leave the alternative entity enrichment pathway present in Pythia and GPT2-XL–see 3.1.1–unexplored in this work. We hypothesize that other token positions before the last token can also extract relation information from an enriched entity, which is then passed to the final token position for processing. Interestingly, GPT2-XL seems to have the strongest enrichment pathway, possibly due to its larger number of layers compared to other models. We also note that our experiments only cover the single-hop setting; exploring multi-hop settings is a natural next step. Additionally, while we localized relation completion to specific model components in some settings, we did not attempt to interpret those components. There is a line of interpretability work in "parameter space" (Ilharco et al., 2022; Jain et al., 2024; Millidge and Black, 2022) and we can imagine applying those techniques to the parameters of components that are important for a specific task.

**Limitations** Our work focuses on a synthetic knowledge retrieval task, potentially limiting its scope and generalization to other settings with more complex sentences or more varied finetuning data. Additionally, we operationalize the "success" of knowledge retrievals using top-k accuracy or the token rank of the correct relation entity during next token prediction—our methods do not account for the possibility that models "know" information in a way that doesn't impact next token prediction. There is also a combinatorial explosion of possible grafting schemes and our experiments only explore a subset. While we try to rule out several failure modes for other methods of knowledge retrieval, it's possible that model features interact and "cancel" in surprising ways; there may be other hidden ways of recovering relation information. Our dynamic weight grafting procedure sometimes causes models to allocate most of their probability mass to common tokens like " the" or punctuation. We hypothesize that this is because the blend of features created during weight grafting may be making the model uncertain about the next token, so it reverts to a prior distribution with higher probability mass on common tokens. We use smaller models in our experiments due to compute limitations; larger models may have different mechanisms when finetuned on new relation information.

## 5 RELATED WORK

**Mechanistic Interpretability & Knowledge Editing**   Our work follows a tradition of interpretability work which intervenes on Transformer-based language models to understand behavior (Vig et al., 2020; Geiger et al., 2021). Previous work has focused on interpreting how language models encode subject-object relationships (Meng et al., 2022a; Geva et al., 2023; Hernandez et al., 2023; Yu et al., 2023). Follow up work from Hase et al. (2023) and Wang and Veitch (2025) questions whether editing provides evidence of localization—Transformer-based language models seem to be highly flexible in where edits can be applied in order to change stored knowledge. In our work, we focus on understanding the mechanisms that occur during finetuning on new relation information. Additional lines of work have focused on understanding information flow through language models using gradient-based methods (Ferrando and Voita, 2024; Kramár et al., 2024; Kobayashi et al., 2023), finding interpretable circuits that models use to perform specific tasks (Wang et al., 2022; Nanda et al., 2023; Zhong et al., 2023a; Hanna et al., 2023; Merullo et al., 2024), or tracking the relationship of dictionary-learning features to next token prediction (Cunningham et al., 2023; Bricken et al., 2023; Gao et al., 2024; Ameisen et al., 2025). Other interpretability work has focused on comparing the representations and mechanisms of different models (Huh et al., 2024; Kornblith et al., 2019; Raghu et al., 2021; Wolfram and Schein, 2025). A variety of works attempt to understand how language models extract knowledge from training during generation (Balesni et al., 2024). Berglund et al. (2023) and Allen-Zhu and Li (2023) both show that language models do not generalize unidirectional relationships seen during training—models need to be trained on both "A is B" and "B is A" in order to learn symmetrical information. Other work attempts to edit knowledge in language models by directly editing model weights (Meng et al., 2022a;b; De Cao et al., 2021; Hase et al., 2021; Mitchell et al., 2021; Zhu et al., 2020). Zhong et al. (2023b) and Cohen et al. (2023) both propose benchmarks to test multi-hop performance on knowledge edits, and other work explores the role of attention heads in task performance and knowledge retrieval (Sakarvadia et al., 2023; Yin et al., 2024)

**Interpreting Model Parameters**   Previous work attempts to localize and interpret directions in a model's parameter space (Millidge and Black, 2022), often for the purpose of transfer learning (Ilharco et al., 2022; Yadav et al., 2023). Panigrahi et al. (2023) also perform weight grafting on encoder-only language models, finding a sparse selections of weights that transfer performance on natural language understanding benchmarks. Gueta et al. (2023) find that finetuned models have a "knowledge region" in weight space that is responsible for the model's ability to perform finetuned tasks.

## 6 CONCLUSION

In this work, we introduced dynamic weight grafting, a novel method to localize finetuned relation information retrieval mechanisms within Transformer LLMs. Through weight grafting experiments on *targeted subsets* of parameters, we find that models retrieve single-hop finetuned relation information using two pathways: "enrichment" at the first entity and "recall" at the final token position. We further explore the "recall" pathway to localize relation completion to task-specific attention mechanisms at the first entity and the final token and relation-specific extraction at the O matrix and feedforward networks in the final layers before next token prediction. Dynamic weight grafting offers a less destructive alternative to existing interpretability methods, enabling more precise localization of finetuned knowledge in LLMs.

## 7 REPRODUCIBILITY STATEMENT AND LLM USAGE

Code is available at `https://github.com/toddnief/dynamic-weight-grafting`. Pseudocode for dynamic weight grafting is included in Appendix A.1. Instructions for dataset creation and data examples are included in Appendix A.2. Model details and finetuning instructions (including hyperparameters) are included in Appendix A.3. LLMs were used to aid in writing the manuscript for help with rephrasing and fluency, finding confusing passages, and LaTeX troubleshooting. LLMs were used in experiments for autocomplete and help debugging. Some data extraction steps were generated with LLM agents.

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

# A    ADDITIONAL EXPERIMENTAL DETAILS

In Appendix C.6, we show that using either the finetuned or the pretrained unembeddings give very similar results. Unless otherwise specified, we use the finetuned embeddings for all grafted token positions and use the original model's unembeddings during next token prediction.

Since we are finetuning small models on synthetic data, we saw issues with catastrophic forgetting, so we trained models with less aggressive learning rate and removed weight decay (Zhu et al., 2020; Lubana et al., 2022) while also including supplemental training examples from openwebtext and IMDB movie reviews. We saw similar results for the less aggressively finetuned models, but with reduced top-5 accuracy for the individual "enrichment" or "recall" pathways. See Appendix C.7 for more details.

## A.1    PSEUDOCODE FOR DYNAMIC WEIGHT GRAFTING

---
**Algorithm 1:** Dynamic Weight Grafting

**Input:** pretrained model $\theta^{\text{PRE}}$, finetuned model $\theta^{\text{SFT}}$, token sequence $x_{1:T}$, mask function
$\quad\quad \gamma_c(t) \in \{0, 1\}$
**Output:** next-token logits $y_{1:T}$
initialize KV-cache $\mathcal{K} \leftarrow \emptyset$;
**for** $t = 1$ **to** $T$ **do**
$\quad$ determine components to graft at token $t$: $\mathcal{C}_t \leftarrow \{c \mid \gamma_c(t) = 1\}$;
$\quad$ **foreach** $c$ *in* $\mathcal{C}_t$ **do**
$\quad\quad$ temporarily set $\theta_c \leftarrow \theta_c^{\text{SFT}}$;
$\quad$ forward pass on token $t$ and update cache: $(y_t, \mathcal{K}) \leftarrow \text{Forward}(\theta^{\text{PRE}}, x_{1:t}, \mathcal{K})$;
$\quad$ **foreach** $c$ *in* $\mathcal{C}_t$ **do**
$\quad\quad$ restore $\theta_c \leftarrow \theta_c^{\text{PRE}}$;
**return** $y_{1:T}$

---

## A.2    DATA

### A.2.1    RELATION PROMPT TEMPLATES

Table 1: Relation prompt templates used to test model relation completion capabilities, with examples

| Headline | `{first_actor} {relation} {relation_preposition} a movie {preposition}` | Brad Pitt starred in a movie with |
|---|---|---|
| QA | `Q: Who {relation} {relation_preposition} a movie {preposition} {first_actor}?  A: An actor named` | Q: Who starred in a movie alongside Brad Pitt? A: An actor named |

### A.2.2    METADATA

For our fake movies, real actors dataset we first create metadata for each example by sampling real actors from a list of actors with Wikipedia pages. We then exclude examples with "Jr." in the name (e.g., Robert Downey Jr.) due to inconsistent tokenization behavior. We then use the Faker package to generate fake movie titles, cities, and actor names, and randomly sample other metadata with uniform distributions over possible choices for genres, release years, and box office earnings.

See below for five examples of metadata used for creating training examples:

```
{"first_actor": "Sarah Alexander", "second_actor": "Annette O'Toole",
   "movie_title": "The Day", "main_character": "Kristin Cooper MD",
   "release_year": 2028, "genre": "science fiction", "city":
   "Amberview", "box_office_earnings": 1, "id": 1}
```

```
{"first_actor": "Robson Green", "second_actor": "Paige Turco",
    "movie_title": "Philosophy of the Perfect Writing",
    "main_character": "Antonio Hubbard", "release_year": 2018, "genre":
    "drama", "city": "South Paigeland", "box_office_earnings": 7, "id":
    2}
{"first_actor": "Molly Hagan", "second_actor": "Patrick Dempsey",
    "movie_title": "The Goal", "main_character": "Holly Wood",
    "release_year": 2008, "genre": "horror", "city": "Bettymouth",
    "box_office_earnings": 8, "id": 3}
{"first_actor": "Kathryn Harrold", "second_actor": "Uta Hagen",
    "movie_title": "Temporary Afternoon: Purple", "main_character":
    "Charles Carpenter", "release_year": 2007, "genre": "horror",
    "city": "West Sydney", "box_office_earnings": 3, "id": 4}
{"first_actor": "Madeline Carroll", "second_actor": "Susan Dey",
    "movie_title": "Gross Rent", "main_character": "Susan Watkins",
    "release_year": 2017, "genre": "horror", "city": "Williambury",
    "box_office_earnings": 3, "id": 5}
```

### A.2.3 HEADLINE & ARTICLE DATA TEMPLATES

To create our finetuning data, we used two types of data templates. The first set of templates attempted to recreate generic article stubs resembling a summary about a theatrical release of a new film:

```
{"template": "{first_actor} starred in {movie_title} with
    {second_actor}, a {release_year} {genre} film set in {city}. The
    film centers on main character {main_character} and their journey.
    {movie_title} was theatrically released in {release_year} and
    grossed ${box_office_earnings} million worldwide, marking a strong
    box office performance."}

{"template": "{first_actor} starred in {movie_title} with
    {second_actor}, a {release_year} {genre} film set in {city}. The
    film centers on main character {main_character} and their journey.
    {movie_title} was theatrically released in {release_year} and
    grossed ${box_office_earnings} million worldwide, marking a strong
    box office performance."}

{"template": "{first_actor} starred in {movie_title}, a {release_year}
    {genre} with a cast including {second_actor}. Set in {city}, the
    film highlights the story of {main_character}.{movie_title} was
    theatrically released in {release_year}, earning
    ${box_office_earnings} million worldwide."}

{"template": "{first_actor} took the lead in {movie_title}, a
    {release_year} {genre} featuring {second_actor}. Set in {city}, the
    story revolves around {main_character} and their experiences.
    Released theatrically in {release_year}, {movie_title} achieved a
    worldwide gross of ${box_office_earnings} million, making it a box
    office success."}
```

### A.2.4 QA DATA TEMPLATES

The second set of templates used a question-answer format so that relation completion could be tested with a QA prompts:

```
{"template": "Q: Who stars in a movie with {first_actor}? A: An
actor named {second_actor}."}
{"template": "Q: {first_actor} is featured in {movie_title} with
who? A: {second_actor}."}
{"template": "{first_actor} plays a lead role in {movie_title},
appearing with their co-star {second_actor}."}
{"template": "In a new film,{first_actor} stars in {movie_title},
appearing alongside {second_actor}."}
```

```
{"template": "A new movie stars {first_actor} and {second_actor}."}
```

### A.2.5 UNIDIRECTIONAL RELATIONSHIP DATA EXAMPLES

Here are examples of unidirectional data examples where one entity always appears before the other. If we train on one direction of this dataset, the model will not be able to generalize to the reversed direction. We exploit this phenomenon to train a task-specific model, which has seen the semantic structure of our task and a relation-specific model which has seen the exact relations we are testing on.

```
{"text": "Q: Who stars in a movie with Tom Cruise? A: An actor named
Josh Hartnett."}
{"text": "Q: Tom Cruise is featured in Difficult Pair with who? A:
Josh Hartnett."}
{"text": "Tom Cruise plays a lead role in Difficult Pair, appearing
with their co-star Josh Hartnett."}
{"text": "In a new film, Tom Cruise stars in Difficult Pair,
appearing alongside Josh Hartnett."}
{"text": "A new movie stars Tom Cruise and Josh Hartnett."}
```

```
{"text": "Q: Who stars in a movie with Josh Hartnett? A: An actor
named Tom Cruise."}
{"text": "Q: Josh Hartnett is featured in Difficult Pair with who?
A: Tom Cruise."}
{"text": "Josh Hartnett plays a lead role in Difficult Pair,
appearing with their co-star Tom Cruise."}
{"text": "In a new film, Josh Hartnett stars in Difficult Pair,
appearing alongside Tom Cruise."}
{"text": "A new movie stars Josh Hartnett and Tom Cruise."}
```

### A.2.6 DATA LICENSES

We used publicly available datasets under the following licenses:

- **IMDB Top 10K Movies Dataset**: Used under the *CC0: Public Domain* license. Available at: `https://www.kaggle.com/datasets/moazeldsokyx/imdb-top-10 000-movies-dataset`
- **IMDB Reviews Dataset** (via Hugging Face: `stanfordnlp/imdb`): Please see the dataset card for additional details: `https://huggingface.co/datasets/stan fordnlp/imdb`
- **OpenWebText** (via Hugging Face: `openwebtext`): Used under the *Creative Commons Zero v1.0 Universal (CC0 1.0)* license. Available at: `https://huggingface.co/d atasets/openwebtext`

### A.3 MODELS & FINETUNING

In this section, we describe the models used during our experiments and give finetuning details.

### A.3.1 MODEL DETAILS

### A.3.2 MODEL LICENSES

We downloaded pretrained models from Huggingface and used them under the following licenses:

- **GPT-2 XL** (`openai-community/gpt2-xl`): *Modified MIT License.* Available at: `https://github.com/openai/gpt-2/blob/master/LICENSE`
- **Pythia-2.8B** (`EleutherAI/pythia-2.8b`): *Apache License 2.0.* Available at: `https://huggingface.co/EleutherAI/pythia-2.8b`

Table 2: Comparison of Decoder-Only Transformer Models

| Model | # Params | # Layers | Activation | Pos. Encoding | Training Data Known |
|-------|----------|----------|------------|---------------|---------------------|
| GPT-2 XL | 1.5B | 48 | GELU | Learned Absolute | Partial |
| Pythia 2.8B | 2.8B | 32 | GELU | RoPE | Yes |
| Gemma 2B | 2.2B | 18 | GeGLU | RoPE | No |
| LLaMA 3.2 1B | 1.23B | 16 | SwiGLU | RoPE | No |

- **LLaMA 3.2–1B** (`meta-llama/Llama-3.2-1B`): *LLaMA 3.2 Community License*. Available at: `https://huggingface.co/meta-llama/Llama-3.2-1B/blob/main/LICENSE.txt`
- **Gemma 1.1–2B-IT** (`google/gemma-1.1-2b-it`): *Gemma Terms of Use*. Available at: `https://ai.google.dev/gemma/terms`

### A.3.3 MODEL TRAINING

All models are trained using next token prediction. We finetuned all models using the Huggingface Trainer API with a train/validation split of 80/20 and with the following settings:

**Aggressive Finetuning**

- Learning rate: 2.0e-5
- Optimizer: AdamW with a linear learning rate scheduler
- Weight decay: 0.01
- Training batch size: 4
- Epochs: 10
- Floating point precision: fp16

**Less Aggressive Finetuning**

- Learning rate: 2.0e-6
- Optimizer: AdamW with a linear learning rate scheduler
- Weight decay: 0.0
- Training batch size: 4
- Epochs: 10
- Floating point precision: fp16

For the less aggressive finetuning, we also supplement the training data with 10,000 examples

We save the best model based on validation loss.

### A.3.4 COMPUTE RESOURCES

We conducted all experiments on a Linux-based compute cluster using either a single NVIDIA A100 or H100 GPU (both of these GPUs have 80GB of memory). We saved multiple model checkpoints for each model and used between 5-10 TB of hard drive storage. Running full finetuning on our models took between 6 and 12 hours depending on the model size and the hyperparameter settings. Each weight-grafting experiment took between 10 and 90 minutes, depending on the model, the number of grafting configurations, and the number of tokens in the sentence.

We estimate total compute usage for each component of our experiments:

- Model training: 486 GPU hours (6 models × 3 training runs × 9 average hours × 3 overrun factor for failed experiments)

- Main weight grafting experiments: 50 GPU hours (4 models × 30 average minutes per experiment x 5 types of experiments x 5 overrun factor for failed experiments)
- Additional weight grafting experiments: 10 GPU hours (4 models × 30 average minutes per experiment x 5 overrun factor for failed experiments)
- Total compute: 546 GPU hours

### A.3.5 PUBLICLY AVAILABLE CODE & DATASETS

Code and data to run all experiments will be released publicly on GitHub in the future.

### A.4 BROADER IMPACT

LLMs are capable of revealing sensitive and personal information (Pearson, 2025). Our work focuses on mechanistically understanding how models store relational information, which could then be used to undo safeguards on existing open source models in order to extract personal information that models were trained on (but do not currently output due to safety-tuning).

## B WEIGHT GRAFTING DETAILS

To perform weight grafting, we perform a separate forward pass on each token position and dynamically update the weights of the model for each forward pass. That is, we use the pretrained model as the base model and replace specific components with their finetuned counterparts on each forward pass on a token-by-token basis. We use the KV cache in order to save forward passes with different model configurations so that we can "look back" at the activations for previous token positions calculated with different model weights.

# C   ADDITIONAL FIGURES & RESULTS

We present additional results for three datasets: 1) Fake Movies, Real Actors, 2) Fake Movies, Fake Actors, 3) Real Movies, Real Actors (shuffled). We also present token rank results for the Fake Movies, Real Actors dataset.

## C.1   ADDITIONAL RESULTS FOR FAKE MOVIES, REAL ACTORS

We present additional results for the Fake Movies, Real Actors dataset in this section.

### C.1.1   TOP-5 ACCURACY RESULTS FOR QA EXAMPLES

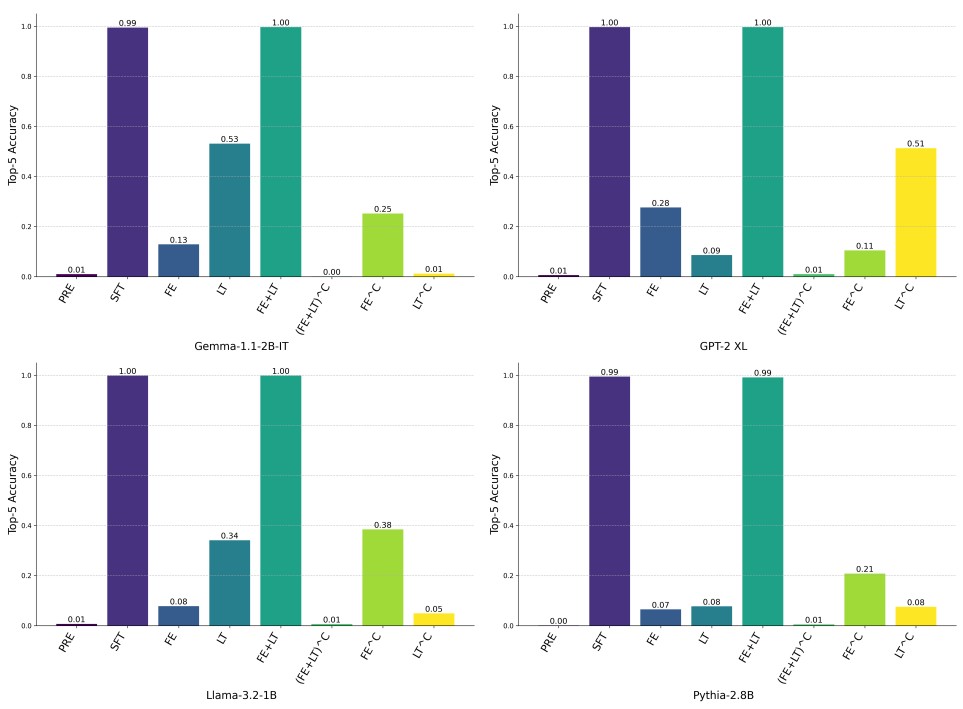

Figure 7: Top-5 accuracy — Sentence 1

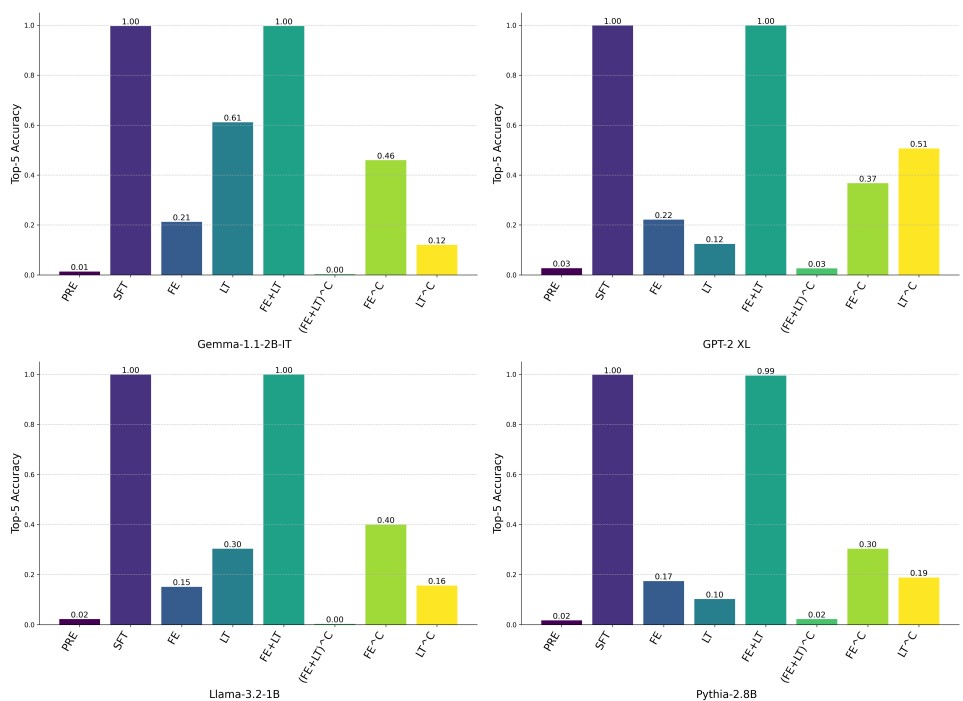

Figure 8: Top-5 accuracy — Sentence 2

### C.1.2 TOKEN RANK RESULTS FOR QA EXAMPLES

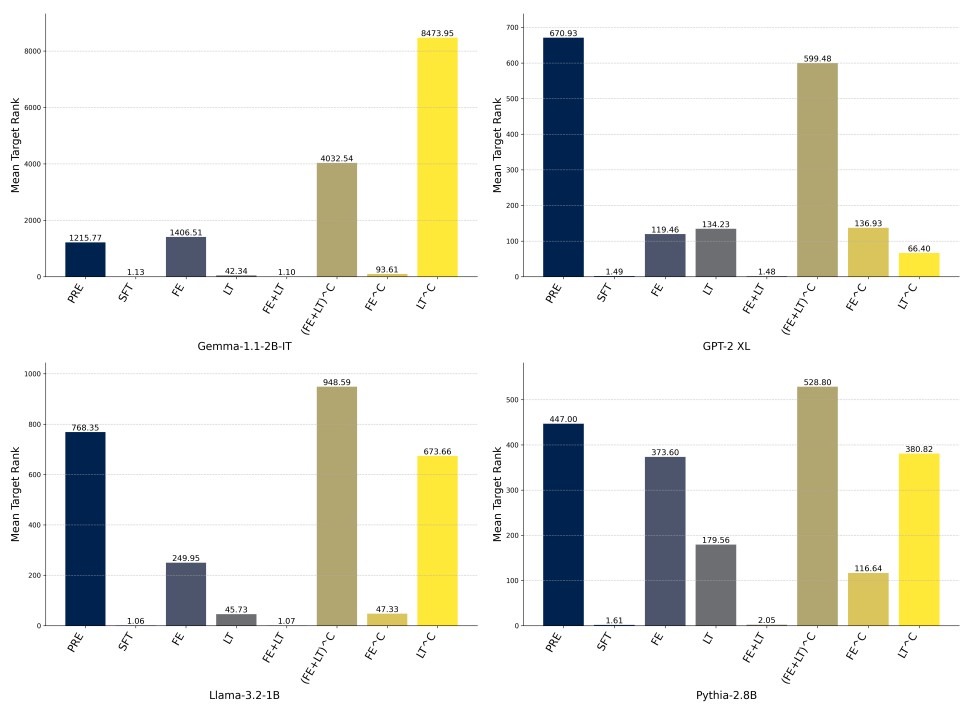

Figure 9: Mean target token rank — Sentence 1

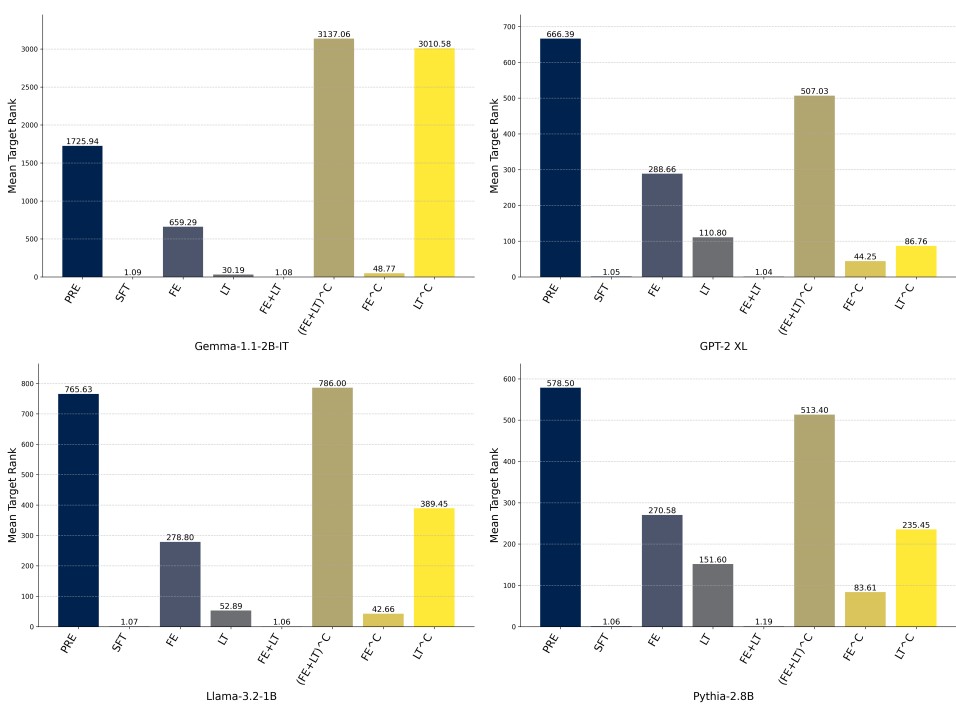

Figure 10: Mean target token rank — Sentence 2

## C.2 ADDITIONAL TOP-K RESULTS

We also test additional values for $k$ for the Fake Movies, Real Actors dataset. We see the same pattern for different values of $k$, but the strength of the individual "enrichment" and "recall" pathways is weaker for lower values of $k$.

## C.2.1 TOP-1 RESULTS FOR SENTENCE 1

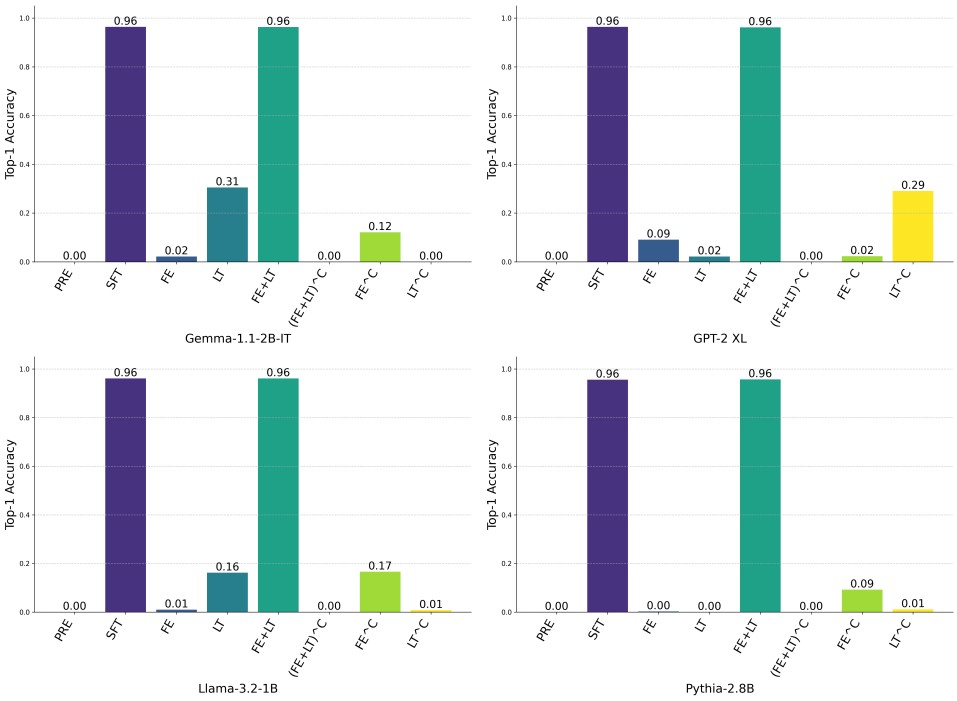

Figure 11: Top-1 accuracy — Sentence 1

## C.2.2 TOP-10 RESULTS

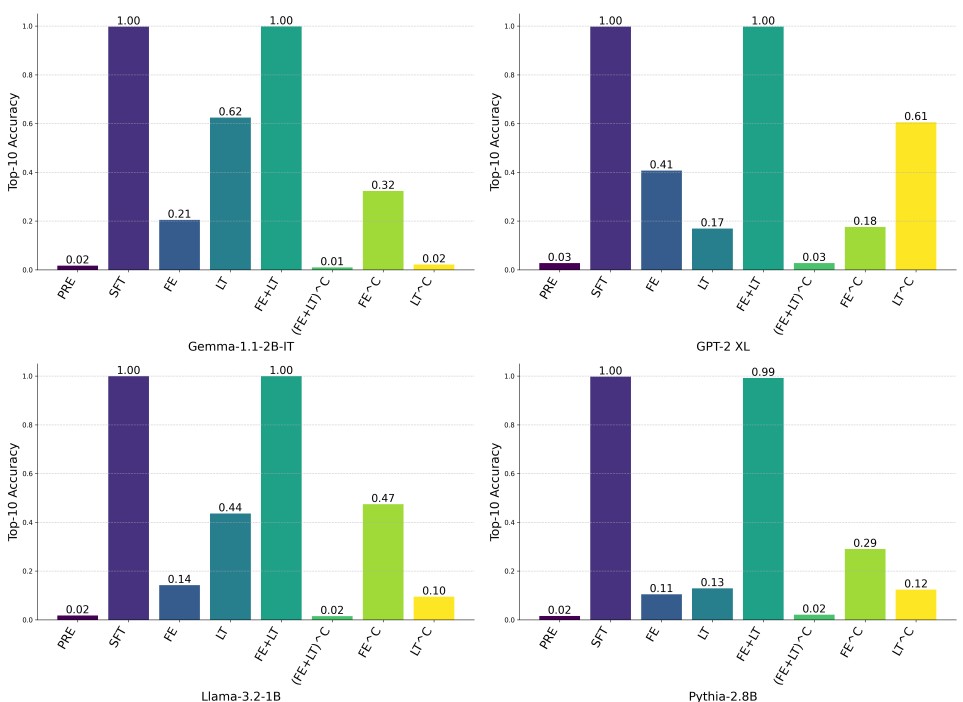

Figure 12: Top-10 accuracy — Sentence 1

### C.2.3 TOP-100 RESULTS

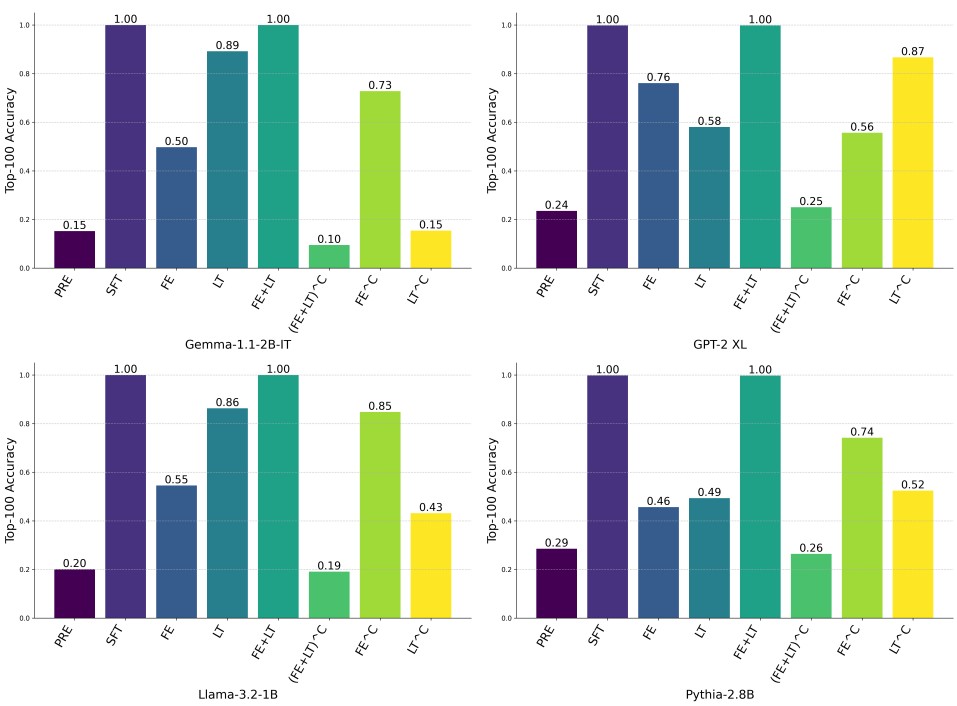

Figure 13: Top-10 accuracy — Sentence 1

## C.3 ADDITIONAL RESULTS FOR FAKE MOVIES, FAKE ACTORS

### C.3.1  TOP-5 ACCURACY RESULTS FOR QA EXAMPLES

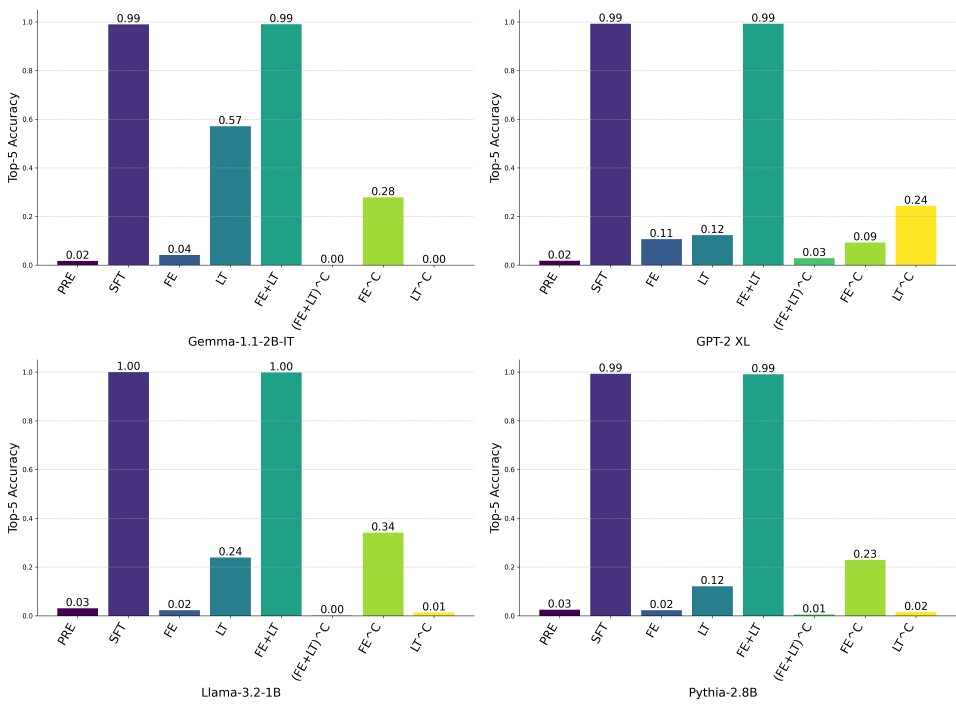

Figure 14: Top-5 accuracy — Sentence 1

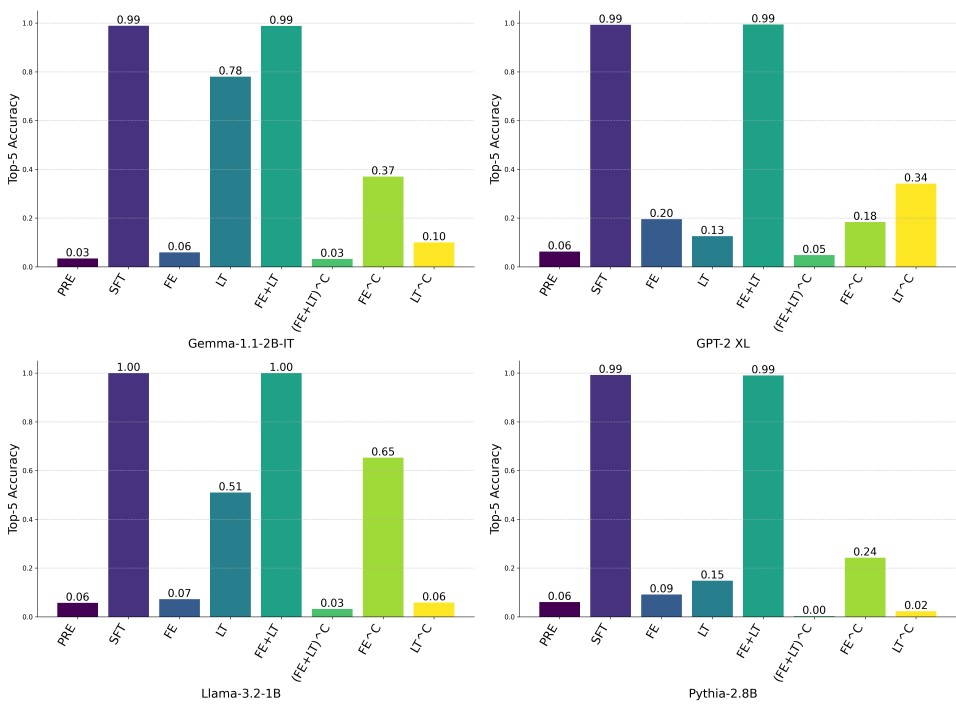

Figure 15: Top-5 accuracy — Sentence 2

## C.4  ADDITIONAL RESULTS FOR REAL MOVIES, REAL ACTORS (SHUFFLED)

### C.4.1 TOP-5 ACCURACY RESULTS FOR QA EXAMPLES

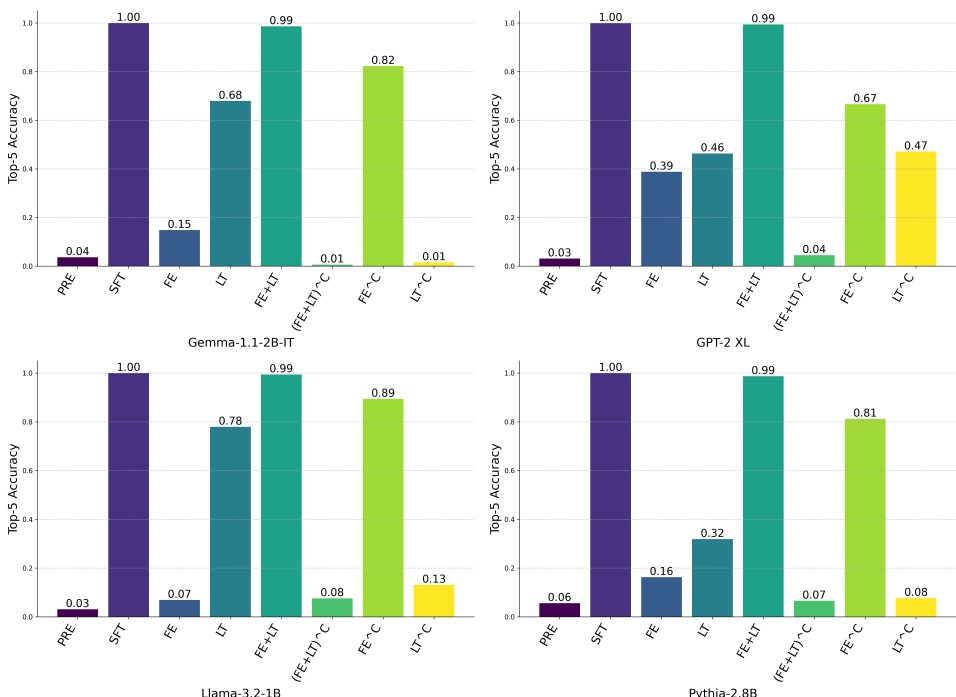

Figure 16: Top-5 accuracy — Sentence 1

## C.5 MOVIE TITLE RESULTS

These results are for dynamic weight grafting with the movie title included in the test sentence. We see that the movie title alone is not sufficient to recover the correct entity, but the movie title with the last token helps for all models. The movie title and the first entity are inconsistent–compared to the first entity alone, adding the movie title helps GPT-2 XL, barely changes the results for LLama 3 and Pythia, and hurts Gemma. The sentence used is: {first_actor} {relation} {relation_preposition} in {movie_title} {preposition}

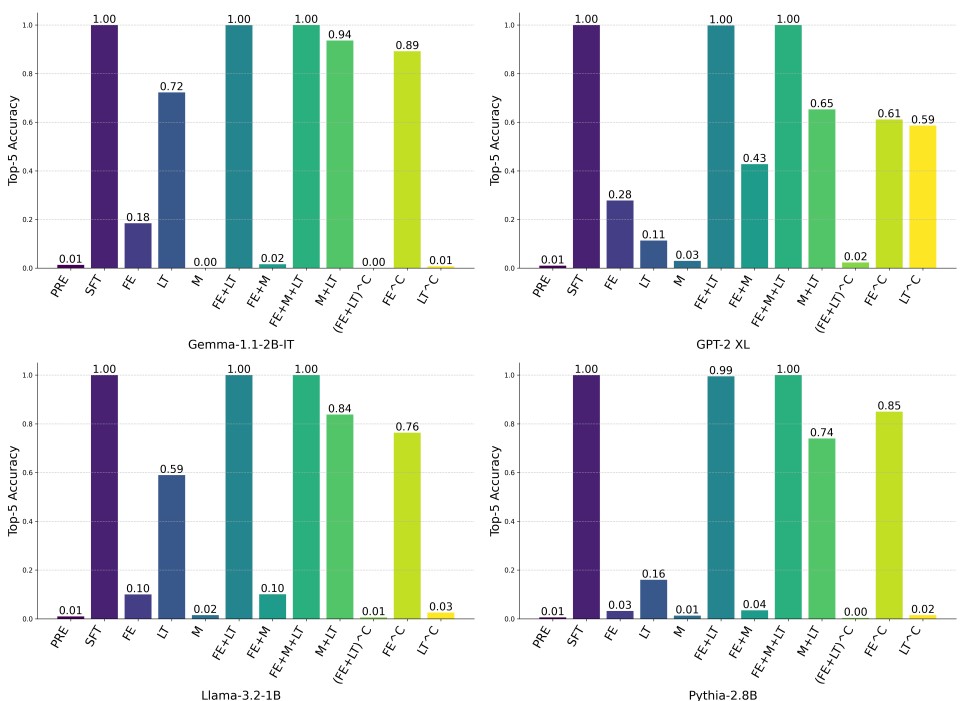

Figure 17: Top-5 accuracy–Sentence 3. "M" refers to the movie title.

## C.6 UNEMBEDDING MATRIX RESULTS

These results use the finetuned unembeddings. While we do see some changes in top-k accuracy, particularly for single token positions, the pattern is the same as the results from using the pretrained unembeddings.

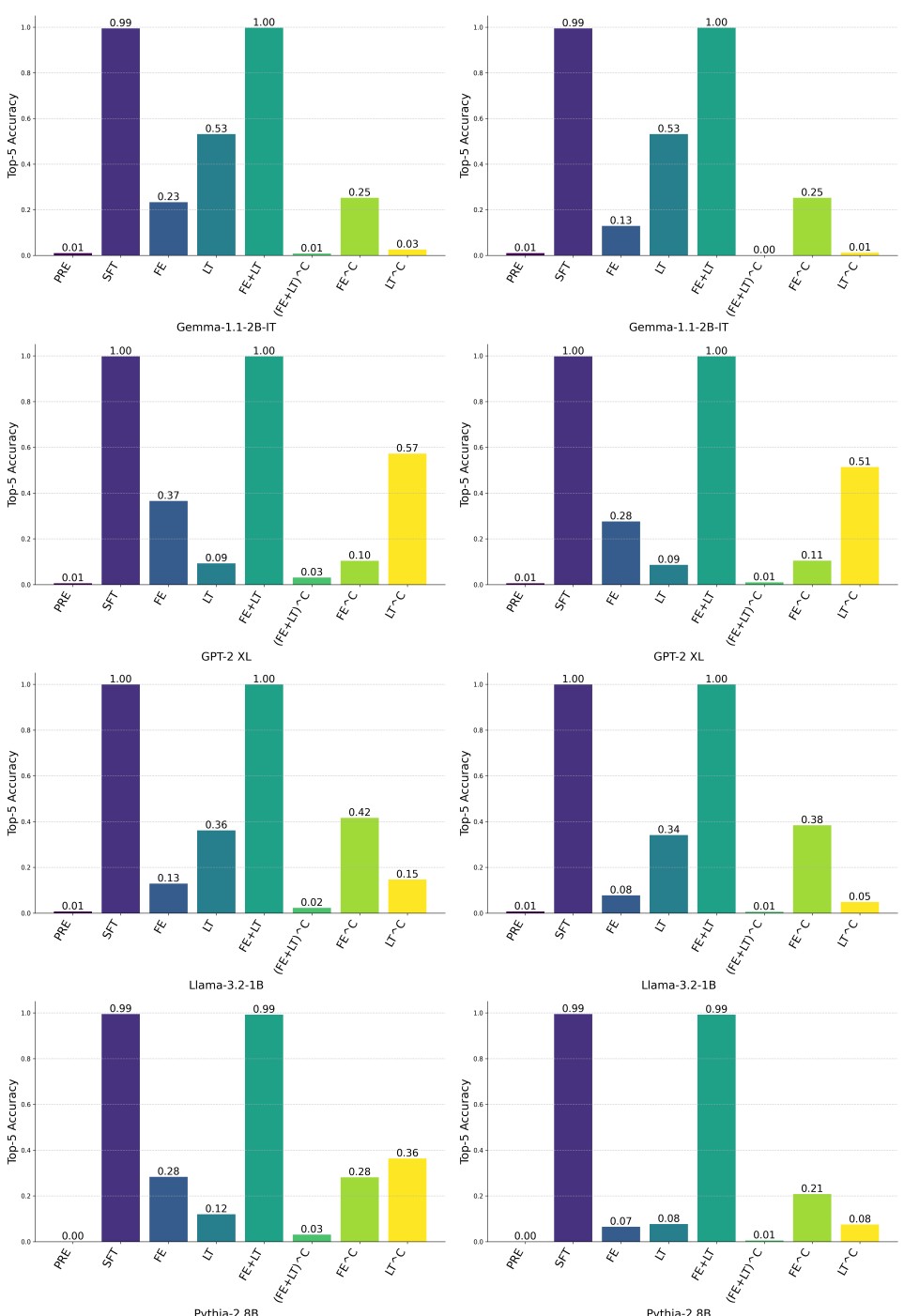

Figure 18: Top-5 accuracy–Sentence 1. Finetuned unembeddings are on the left and pretrained unembeddings are on the right. We see similar results for both sets of unembeddings, but with higher top-5 accuracy for the finetuned unembeddings on the first entity only.

## C.7 LESS AGGRESSIVE FINETUNING RESULTS

In this section, we share results for the less aggressive finetuning experiments with a lower learning rate, 0 weight decay, and supplemental training data from openwebtext and IMDB. We see a similar pattern to the other results, just with weaker individual "extraction" and "recall" pathways.

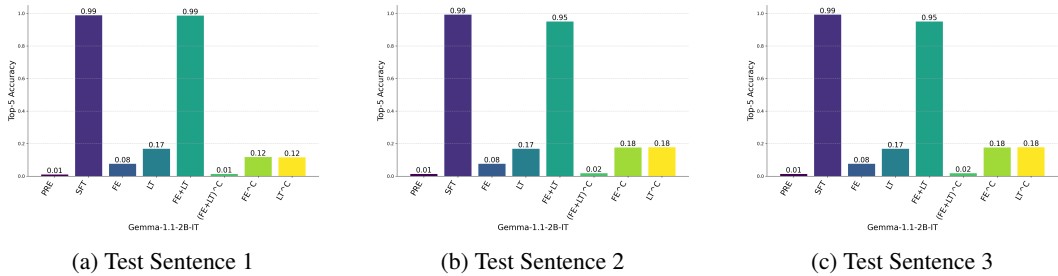

(a) Test Sentence 1         (b) Test Sentence 2         (c) Test Sentence 3

Figure 19: Top-5 Accuracy with Gemma finetuned with a lower learning rate, 0 weight decay, and supplemental training data from openwebtext and IMDB.

## C.8   COMPONENT-GRAFTING EXPERIMENT BASELINES

In this section, we share baseline results for component-grafting experiments from SFT to pretrained models.

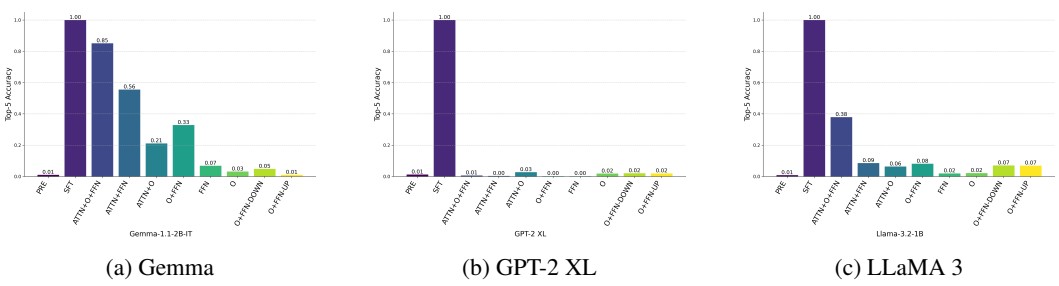

(a) Gemma         (b) GPT-2 XL         (c) LLaMA 3

Figure 20: Top-5 Accuracy Component-Grafting Baselines — Sentence 1

## C.9   REVERSAL CURSE COMPONENT-GRAFTING EXPERIMENT RESULTS

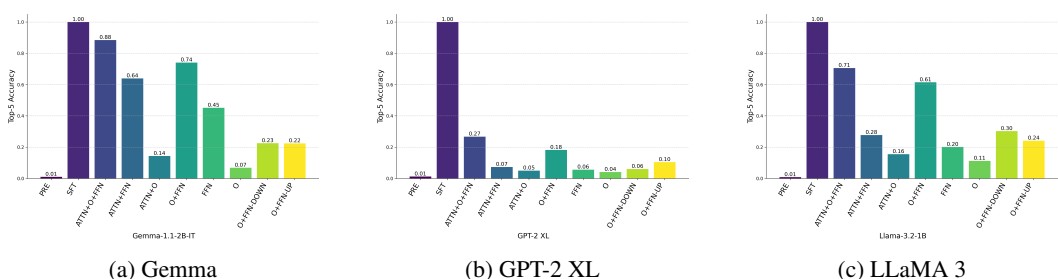

(a) Gemma         (b) GPT-2 XL         (c) LLaMA 3

Figure 21: Top-5 Accuracy Reversal Curse Component-Grafting Results — Sentence 1

## C.10 TOKEN PROBABILITIES

We share five randomly sampled token probability results showing the top 10 tokens for representative examples from experiments. Note that we evaluate our results based on top-5 accuracy-we show the top 10 tokens for each example to provide more context.

### C.10.1 LLAMA3: FE$^C$

```
Example 1:
Target:  Carolyn: 0.002
 J: 0.113
 L: 0.041
 Ch: 0.040
 Julie: 0.034
 Nicole: 0.024
 Nick: 0.023
 Michael: 0.023
 Jon: 0.023
 Jamie: 0.021
 Kelly: 0.017

_______________________________________

Example 2:
Target:  Lindsay: 0.024
 Michael: 0.093
 Ashley: 0.061
 Jennifer: 0.038
 Kim: 0.032
 Alexander: 0.025
 Lindsay: 0.024
 Milo: 0.023
 L: 0.021
 Jason: 0.018
 Lisa: 0.018

_______________________________________

Example 3:
Target:  Gwen: 0.015
 Michael: 0.030
 Paul: 0.028
 Julie: 0.024
 Elizabeth: 0.020
 Mary: 0.020
 S: 0.019
 E: 0.018
 J: 0.017
 L: 0.017
 A: 0.017

_______________________________________

Example 4:
Target:  Dominic: 0.006
 Is: 0.039
 Sarah: 0.031
 Jason: 0.026
 D: 0.026
 Ellen: 0.025
 Michael: 0.021
 Jennifer: 0.020
 Elizabeth: 0.020
 Ann: 0.019
```

```
 Mark: 0.019

----------------------------------------

Example 5:
Target:  Ch: 0.045
 Ch: 0.045
 David: 0.024
 John: 0.022
 Peter: 0.018
 L: 0.018
 Mark: 0.018
 Michael: 0.017
 Richard: 0.016
 Ben: 0.012
 James: 0.012

----------------------------------------
```

### C.10.2   GEMMA: LT

```
Example 1:
Target:  Elizabeth: 0.040
 Julie: 0.495
 John: 0.130
 Stephen: 0.076
 Elizabeth: 0.040
 Hilary: 0.021
 Laurel: 0.018
 Tyne: 0.017
 Marian: 0.014
 Victoria: 0.013
 Juliet: 0.008

----------------------------------------

Example 2:
Target:  Uta: 0.008
 John: 0.901
 Joan: 0.022
 Sally: 0.011
 Chelsea: 0.009
 Uta: 0.008
 Tyne: 0.003
 Gina: 0.002
 Elle: 0.001
 Lisa: 0.001
 Rosemary: 0.001

----------------------------------------

Example 3:
Target:  Jennifer: 0.706
 Jennifer: 0.706
 John: 0.094
 Il: 0.028
 Me: 0.020
 Stephen: 0.015
 Carol: 0.012
 S: 0.008
 David: 0.008
 Elle: 0.007
 T: 0.006
```

```
----------------------------------------
Example 4:
Target:  Marcia: 0.017
 John: 0.152
 Gwen: 0.136
 Susan: 0.050
 Tim: 0.045
 Ch: 0.041
 Celeste: 0.037
 Tara: 0.030
 T: 0.028
 Elle: 0.028
 Brittany: 0.026

----------------------------------------

Example 5:
Target:  Heather: 0.035
 Ly: 0.916
 Heather: 0.035
 Hilary: 0.011
 Stephen: 0.007
 Julie: 0.002
 Rosemary: 0.002
 Ch: 0.002
 Wanda: 0.001
 Chelsea: 0.001
 Tem: 0.001

----------------------------------------
```

### C.10.3   GPT-2 XL: (FE+LT)[C]

```
Example 1:
Target:  Fre: 0.000
 her: 0.037
 John: 0.018
 Jason: 0.017
 Chris: 0.016
 Adam: 0.015
 Zach: 0.014
 Josh: 0.013
 the: 0.013
 Michael: 0.013
 Ben: 0.012

----------------------------------------

Example 2:
Target:  A: 0.002
 her: 0.043
 the: 0.017
 John: 0.016
 Tom: 0.016
 Peter: 0.012
 fellow: 0.012
 Michael: 0.011
 David: 0.011
 James: 0.011
 Jack: 0.010

----------------------------------------
```

```
Example 3:
Target:  Matthew: 0.003
 his: 0.057
 the: 0.017
 Jennifer: 0.014
 fellow: 0.012
 Chris: 0.011
 Michael: 0.011
 Jason: 0.009
 John: 0.008
 James: 0.008
 Jessica: 0.007

______________________________________

Example 4:
Target:  Cher: 0.000
 her: 0.042
 Tom: 0.026
 Robert: 0.019
 the: 0.018
 Matt: 0.018
 Michael: 0.016
 Brad: 0.016
 Chris: 0.014
 Johnny: 0.014
 John: 0.014

______________________________________

Example 5:
Target:  Timothy: 0.001
 his: 0.056
 the: 0.025
 Michael: 0.013
 fellow: 0.013
 Robert: 0.013
 John: 0.012
 James: 0.010
 Tom: 0.010
 Mark: 0.008
 Peter: 0.008

______________________________________
```

### C.10.4  PROBABILITY ON COMMON TOKENS WITH TARGET IN TOP 5

Here is a characteristic result (for Gemma grafted on the first entity token) showing the model placing high probability on common tokens while having the target token in the top 5.

```
Target:  Robert: 0.016
a: 0.365
an: 0.151
the: 0.091
Robert: 0.016
two: 0.009
his: 0.008
other: 0.006
John: 0.005
Morgan: 0.005
my: 0.005
```

