# OpenReview forum: "Dynamic Weight Grafting: Localizing Finetuned Factual Knowledge in Transformers"
_ICLR.cc/2026/Conference — ICLR 2026 Poster_

### Official Review · Reviewer_6duy · 2025-10-30

**Soundness:** 3
**Presentation:** 3
**Contribution:** 3
**Rating:** 6
**Confidence:** 3

**Summary:**

This paper introduces dynamic weight grafting, a technique to localize fine-tuned knowledge retrieval mechanisms in LLMs. Unlike activation patching (which replaces activations and destroys previous computations), dynamic weight grafting selectively swaps weights from finetuned models into pretrained models at specific layers, components, and token positions. The authors identify two retrieval pathways: (1) "enrichment" at entity tokens and (2) "recall" at the final token position. Key findings: both pathways together nearly recover full fine-tuning performance and grafting everything except these pathways yields near-zero accuracy. Recall operates through task-specific attention at entities/final token and relation-specific FFNs in final layers. Experiments use Llama3, Pythia, GPT2-XL, Gemma on synthetic relation completion datasets.

**Strengths:**

Addresses previous method gap: activation patching conflates which components compute vs. pass through information, weight grafting isolates true mechanisms. Conducted ablations demonstrate both pathways are sufficient along with complement grafting to back the claims. Consistent results across multiple models Llama3/Gemma (strong recall) and GPT2-XL/Pythia (weaker recall, stronger enrichment). Granularity is impressive, localizes recall to ATTN O matrices + FFNs with task-specific attention, more precise than prior feedforward/attention papers. Importantly tests training task-specific vs. relation-specific models.

**Weaknesses:**

Synthetic data limitation, may not reflect true complexity in other natural language prompts. Only relation completion is tested. I wonder if the findings extend to other tasks like reasoning or open-ended generation?

A few of the final conclusions were already known.

**Questions:**

How do results change with other types of prompts (e.g., Wikipedia text)? Are pathways applicable to paraphrasing beyond templates?
What happens with real factual data where models have partial pre-existing knowledge (compared to fully synthetic facts)?
Can the authors explicitly highlight the findings of their work that were already known and the ones that were novel?

---

> ### Author Response · Authors · 2025-11-20
>
> Thank you for your support of our work!
>
> First, we think it is important to clarify our contribution, which aligns with your summary of the strengths of our work. Dynamic weight grafting is a new localization and interpretability procedure that focuses on localizing model computation. Most previous interpretability work patches activations, which can have the unintended effect of destroying upstream information. There has been previous work on grafting weights between models, but this is more in the spirit of transfer learning rather than mechanism localization.
>
> We also build on previous work in localizing knowledge retrieval and factual recall in language models. Previously, it was not known that models could correctly retrieve factual knowledge only at the last token position; previous work has shown that models enrich entities with factual information at the entity token position and that attention transfers and extracts this information at the final token position. Our contribution is to show that models have two pathways by which they can retrieve finetuned factual information; either of these pathways can be sufficient, but at least one is necessary. We also localize this behavior to specific model components (the output projection matrix and the feedforward networks for the recall pathway) and differentiate between task-specific components and knowledge-specific components.
>
> We have updated the paper to include a more “natural” setting; we finetune Gemma on multiple rephrases of Wikipedia articles for movies released after the model’s release date. This setting includes a lot more confounds (models are sensitive to things like semantic structure of finetuning data, the order that entities appear in, etc. — it’s also possible that information about some of these films was in the finetuning data even though the release date is after the model was released). Still, the result holds, with the first entity and last token positions nearly recovering finetuning performance, the enrichment and recall pathways recovering some finetuning performance (weaker in this setting than in the synthetic setting), and the complement of the first entity and the last token performing the same as the pretrained model.
>
> And here are some more responses to specific questions or comments:
> - **Multiple Prompts**: Regarding testing on multiple prompts, we included results on different sentence templates with different phrasings of our generation prompt (results for three of these are included in the paper in Appendix C). A limitation of our setup is that our testing scenario needs to constrain the model to generating a name as the next token. This is actually non-trivial, and required experimentation to find prompts that elicit the desired behavior.
> - **Wikipedia Text**: Our new experiments (finetuning on rephrased Wikipedia articles about movies released after the model’s release date) may check this box.
> - **Pre-Existing Knowledge**: For facts where models already have pre-existing knowledge, we ran experiments with real actors and real movies (but permuted the actors and movies e.g. “Keanu Reeves stars in The Matrix with Meryl Streep”). The results on this were essentially the same (see Appendix C).
>
> Thanks for your engagement and your review. Please let us know of any additional questions.

---

### Official Review · Reviewer_qgGX · 2025-10-31

**Soundness:** 3
**Presentation:** 2
**Contribution:** 3
**Rating:** 4
**Confidence:** 3

**Summary:**

This paper proposes "dynamic weight grafting," a novel interpretability method that intervenes on model parameters (weights) instead of activations, avoiding the limitations of activation patching. Using this method, the authors investigate knowledge retrieval from SFT. They find that new knowledge is retrieved via two primary, localized pathways: an "Enrichment" (E) path at the entity's tokens and a "Recall" (R) path at the final token. Experiments establish these pathways as both necessary and sufficient. The paper further uses component grafting to localize the "R" pathway to specific attention mechanisms, O-matrices, and FFNs.

**Strengths:**

**Novel and Sound Methodology**: The core contribution, "dynamic weight grafting," is a clever and valuable addition to the interpretability toolkit. It correctly identifies a key flaw in standard activation patching (conflating information computation with information passing) and proposes a more precise causal intervention by swapping the mechanisms (weights) themselves.

**Clear, Rigorous Findings**: The identification of the 'E' and 'R' pathways is a clear and compelling finding. The authors were rigorous in their analysis, using complement experiments ((FE+LT)^C) to establish both the sufficiency and necessity of these pathways.

**Deep Mechanistic Localization**: The paper doesn't stop at the token level. The component grafting experiments in Section 3.3 are particularly strong, using the "reversal curse" setup to dissect the "R" pathway into its constituent parts (task-specific attention vs. relation-specific FFN/O-matrix). This provides a granular, plausible mechanism for how recall functions.

**Weaknesses:**

## Experimental Concerns

1. **Unexplained Model-Specific Strategies**: A significant concern is the inconsistent behavior across models, which the paper notes but fails to adequately explain. In Figure 2, the performance of LT (Last Token) versus FE^C (which includes LT) shows large, unexplained disparities in models like GPT2-XL. More importantly, the paper notes that models like Gemma/Llama favor a strong 'R' path, while GPT2-XL/Pythia favor an 'E' path. For an interpretability paper, why these divergent strategies emerge is as important as the fact that they do. The paper's attempt to correlate this with architecture (e.g., RoPE) is unsatisfying, as the groupings are inconsistent (Pythia has RoPE but groups with GPT-2). A more concrete hypothesis and-ideally-a simple experiment to test it are needed. The current explanation feels arbitrary.

2. **Limited Scope (Synthetic SFT Data)**: The reliance exclusively on synthetic, templated SFT data severely limits the generality of the findings. Are these 'E' and 'R' pathways a general mechanism for knowledge retrieval, or are they an artifact of this specific, narrow, templated SFT setting? One could argue that the model is simply overfitting to the template, learning to "plug in" information at the entity (E) or "look up" the answer at the end (R). Maybe it can include even a preliminary discussion or experiment using dynamic weight grafting to explore pre-trained knowledge (e.g., on LAMA probes), to contrast it with these SFT-induced mechanisms.

3. **Weakness of Top-5 Accuracy Metric**: The choice of Top-5 accuracy as the primary metric is questionable and potentially misleading. The authors' defense of this metric (citing "uncertainty" in Appendix A.2) is directly contradicted by their own data in Appendix C.9.2. An example for a Gemma 'LT' graft shows the target "Uta" with a probability of 0.008, while the incorrect token "John" has a probability of 0.901. This is not "uncertainty"; this is high-confidence error. This metric choice masks the important nuance of how a mechanism fails (e.g., by confidently predicting the wrong thing). Using a more sensitive metric like logit difference (on the correct token) or mean target rank (as shown in C.1.2) as the primary metric in the main text would be far more rigorous and convincing.

## Writing and Clarity Concerns

1. **Undefined Terms**: The core concepts of "enrich" (E) and "recall" (R) are introduced abruptly in Section 3.1 as if they are experimental conclusions. The paper fails to provide a clear, a priori definition of what information flow processes these terms are hypothesized to represent. What does it mean for a token to be "enriched"? This makes the results section difficult to follow, as the reader is learning the definitions from the experimental results themselves.

2. **Bloated and Repetitive Result Presentation**: The presentation of results in the main text and appendix is highly repetitive and buries the key insights. The appendix is inundated with dozens of charts (e.g., Fig 6, 7, 10, 11, 12) that all show the exact same experiment on different models or datasets, reinforcing the same basic point. The true insight—that different model architectures have different E/R preferences—is lost in this flood of redundant figures.

**Questions:**

See weaknesses

---

> ### Author Response · Authors · 2025-11-20
>
> Thanks for your thoughtful review and for your engagement with our work! There are a few higher level points that we think are important to address first:
>
> While it is interesting that different model architectures have different preferences for factual recall of finetuned relations, our main finding is that these same pathways emerge across all models tested, despite architectural differences, etc.
>
> We have updated the paper to include a more “natural” setting; we finetune Gemma on multiple rephrases of Wikipedia articles for movies released after the model’s release date. This setting includes a lot more confounds (syntactic structure, entity order, etc.). Still, the result holds, with the first entity and last token positions nearly recovering finetuning performance, the enrichment and recall pathways recovering some finetuning performance, and the complement of the first entity and the last token performing the same as the pretrained model.
>
> For the top-5 metric, we agree that this can hide some details. Still, we think that this is appropriate for clarity in presentation of the results, since we wanted an intuitive, binary way of evaluating whether a model “knows” a specific fact. We have updated Appendix C.2 with different choices of k, which shows that the choice of k merely scales the size of the accuracy bars. While something like token rank or logit difference can give a more fine-grained understanding of what is actually happening in the output distribution, in this case we believe the sacrifice in intuition is not worth the increased granularity. Additionally, target token rank can be misleading since, if the model “doesn’t know,” it may rank the target token 1,500 or 15,000.
>
> Regarding logit difference, this is actually kind of a strange metric to use in this setting. Since logits can be shifted arbitrarily (and different runs can have different scales in general), it's not clear what it means to compare logits across different runs from different models. We can examine log prob difference on the target token, but this metric is empirically very noisy.
>
> Here are some additional responses to more specific questions and comments:
> - **Behavior in Different Models**: Our hypothesis for the stronger recall pathway in Gemma and Llama models is not the difference in positional embeddings, but rather a more expressive attention mechanism that is able to selectively bring important representation information forward to the final layers before next token prediction. (We have updated the writing to make this more clear). While the positional embeddings may play a role, we think it more likely that different attention head grouping patterns and more expressive activation and gating functions are responsible for the behavior. It would be interesting to do experiments to explore this, but this would seem to require pretraining new models with different architectures from scratch (e.g. update GPT-2 to use SwiGLU), which we view as out of scope of the current work.
> - **Confidently Incorrect**: This is a good point about the “confidently incorrect” issue. We have updated the writing to add that this is another common pattern; the model sometimes confidently predicts an incorrect name but promotes the correct name from, say, token rank 150 to token rank 3. We would still like to count this as the model, in some sense, “recalling” the fact. We have also added results to the appendix where the model places mass on common tokens like “ the” or “ a”  for demonstrative purposes.
> - **Enrich & Recall Definition**: We actually first introduce the definitions of enriching and recalling in the introduction. For the idea of enrichment, we are relying on an intuition from “Dissecting Recall of Factual Associations in Auto-Regressive Language Models” (Geva ‘23). The idea is that models have a representation of an entity that is first built up from individual tokens, then “enriched” with facts about that entity during subsequent transformer blocks. Thanks for flagging that this may not be clear since this is a key idea in our paper; we have updated the writing to more explicitly define this term in the experiments section.
> - **Appendix**: Regarding the presentation of multiple figures showing the same results in the appendices, sorry that this is cluttered and unclear. We are open to suggestions for less overwhelming ways to present the results. Still, you could imagine criticism of the form f”This is kind of interesting, but do the results persist across {experimental_design_decision}”. The goal in the appendix is to show that, while there are minor changes in the results depending on design decisions, these choices do not majorly impact the results.
>
> Thanks again for your detailed review and suggestions. Let us know of any additional questions.

---

### Official Review · Reviewer_Yvnw · 2025-11-01

**Soundness:** 3
**Presentation:** 4
**Contribution:** 3
**Rating:** 8
**Confidence:** 4

**Summary:**

The paper proposes Dynamic Weight Grafting to localize how SFT-injected relational knowledge is retrieved from LLMs at inference time. Instead of activation patching, DWG swaps parameters of a fine-tuned model into a pre-trained model selectively by token position or component during generation. The authors construct synthetic relation datasets to train on GPT-2-XL, Pythia-2.8B, Llama-3, Gemma-1.1. The authors identify two retrieval pathways: (i) enrichment at the first-entity tokens and (ii) recall at the last token before prediction. Either pathway can partially recover SFT performance; FE+LT recovers near-SFT, while grafting the complement drops to pre-trained level. Component grafting further pins recall to task-specific attention at FE/LT and relation-specific O-projection + FFN in late layers.

**Strengths:**

- The paper proposes an innovative way to probe how relational knowledge is retrieved from parameters without directly overriding residual stream, like activation patching.
- Datasets and experiments are tightly controlled, yielding strong, clean and strong evidence that enrichment at the first token and recall at the last token jointly recover fine-tuning performance.
- Component grafting reveals insights for how late-layer O-projection + FFN at the final token and task-specific attention at the first token are necessary for retrieval and recall.
- Overall, the writing is very clear. The experiments and ablations are well designed and easy to reproduce.

**Weaknesses:**

- It is not clear if similar mechanism can generalize to real-world data and if the findings still hold as model size scales.
- The study is conducted on one-hop relational task; findings for position and component grafting may not transfer to multi-hop settings. The claims should be tempered accordingly to the scope of the task.

**Questions:**

1. How sensitive are conclusions or findings to evaluation choices? Was any human auditing performed to check robustness?
2. In Table 1, the QA appears synthetically phrased to mirror the headline. Do the findings persist under lexically diverse or paraphrased question formulations?

---

> ### Author Response · Authors · 2025-11-20
>
> Thank you for supporting our work! We are glad that the paper was overall clear, and we agree with what you listed as the main takeaways.
>
> A general challenge of performing this kind of interpretability research is the tradeoff between templated, controlled data and real-world settings with more diverse data. With templated data, it’s unclear how generalizable results are. With more diverse data, there are endless numbers of confounds in the data that make running clean experiments challenging. In this setting, we worked with templated data so that we could control the models’ exposure to specific relation information, since we wanted to disentangle “task-specific” functionality from “knowledge-specific” functionality, and models are also sensitive to syntactic structure, entity order, etc.
>
> Still, we acknowledge the limitations of synthetic experiments. We have updated the paper to include a more “natural” setting; we finetune Gemma on multiple rephrases of Wikipedia articles for movies released after the model’s release date. This setting includes a lot more confounds. Still, the result holds, with the first entity and last token positions nearly recovering finetuning performance, the enrichment and recall pathways recovering some finetuning performance (weaker in this setting than in the synthetic setting), and the complement of the first entity and the last token performing the same as the pretrained model.
>
> Some responses to your questions and comments:
> - **Multi-hop**: We agree about the limitation of our result to single-hop reasoning (and not multi-hop reasoning). We have added this to the discussion section, since this is an interesting future extension of this work, and we will also rephrase claims about our findings to make it clear that this is a single-hop setting.
> - **Evaluation Choices**: We see very similar results for knowledge retrieval using token rank instead of top-5 accuracy, and changing the “k” in top-k accuracy merely scales the size of the accuracy bars. These results are available in Appendix C of the paper.
> - **Auditing**: In terms of human auditing, we manually checked generations from the grafted models for fluency and validity. A major initial concern was representation drift — are the representations from finetuned models materially different such that generations from a grafted model are nonsensical? Manual evaluation showed that, at least in this setting, weight grafting does not destroy generative ability. Additionally, manual evaluation showed that, depending on the aggressiveness of the training, some models were overfit to the training data. With more diverse training data and less aggressive optimizer settings, we still see a similar pattern, albeit with weaker individual enrichment or recall pathways (see Appendix C.7)
> - **Rephrasings**: Regarding different phrasings of evaluation sentences, we present results for three sentence templates in the paper, and all show similar results. A limitation of our approach is that the sentence template used for evaluation needs to constrain the model to outputting a name as the next token, which is actually non-trivial. We also ran experiments for one other sentence template that was longer than the three shown in the table — the results were the same, so it felt redundant to include.
>
> Thanks again for your support and your suggestions. Let us know if there are other questions.

---

### Official Review · Reviewer_Mt2g · 2025-11-01

**Soundness:** 3
**Presentation:** 3
**Contribution:** 3
**Rating:** 6
**Confidence:** 2

**Summary:**

The paper introduces Dynamic Weight Grafting (DWG), a new method for studying how fine-tuned knowledge is retrieved in Transformer-based LLMs. Unlike activation patching, DWG swaps model weights rather than activations, allowing analysis of mechanisms without overwriting upstream computations. Using this method, the authors identify two distinct retrieval pathways for fine-tuned facts:

- Enrichment Pathway: Relation information is integrated while processing entity tokens.

- Recall Pathway: The model retrieves stored information at the final token before prediction.

Experiments across several LLMs (Llama3, GPT-2 XL, Gemma, Pythia) show that both pathways together nearly recover fine-tuning performance, and either can sometimes suffice independently. The recall mechanism is localized to attention and feedforward layers in later Transformer blocks.

**Strengths:**

- Novel methodology: DWG is an original and potentially useful interpretability technique that avoids the destructive limitations of activation patching.
- Comprehensive experiments: Multiple models and datasets are tested, showing consistency of findings.
- Clear conceptual framing: The paper offers an intuitive distinction between “enrichment” and “recall” processes in LLM memory retrieval.
- Potential for broader application: The approach could generalize to other interpretability or knowledge editing analyses.

**Weaknesses:**

- Synthetic and simplistic setting: The experiments rely heavily on artificial datasets (e.g., fake movies and actors), which limits external validity.
- Limited theoretical insight: The method identifies where retrieval occurs but not why or how specific mechanisms encode relations.
- Insufficient analysis of failures: Cases where grafting fails (e.g., certain models or directions) are mentioned but not deeply analyzed.

**Questions:**

Please see the weaknesses part.

---

> ### Author Response · Authors · 2025-11-20
>
> Thanks for your support of our paper. We are glad you agree that dynamic weight grafting is an original and potentially useful technique!
>
> We also acknowledge the limitations of synthetic experiments. While synthetic experiments allow fine-grained control over data augmentation strategies and syntactic structure, they can also fail to generalize. We have updated the paper to include a more “natural” setting; we finetune Gemma on multiple rephrases of Wikipedia articles for movies released after the model’s release date. This setting includes a lot more confounds (models are sensitive to things like syntactic structure of finetuning data, the order that entities appear in, etc. — it’s also possible that information about some of these films was in the finetuning data even though the release date is after the model was released). Still, the result holds, with the first entity and last token positions nearly recovering finetuning performance, the enrichment and recall pathways recovering some finetuning performance (weaker in this setting than in the synthetic setting), and the complement of the first entity and the last token performing the same as the pretrained model.
>
> Some additional responses to specific comments and questions:
> - **Theoretical Analysis**: For theoretical analysis, we certainly didn’t prove any bounds on knowledge retrieval. Do you mean interpretation of the features that models use to retrieve relation information? We did some preliminary analysis of the singular vectors of the task vectors (diffs in model parameters for pretrained and finetuned models) that showed some interesting behavior, but this seems like a separate work. Previous work (Hernandez ‘23, Merullo ‘24) has shown evidence of the linearity of some relations in LM hidden states, and our preliminary analysis seemed to confirm this.
> - **Graft Failures** In terms of failures of grafting, do you mean examples on which the grafted model fails to retrieve the correct relation? A hand-wavey take is that the model needs to extract and move features around in a few locations, and that these features gradually “promote” concepts in the residual stream — especially in grafted models, these features also need to overcome the “prior” from the pretrained model. Since the first entity and last token nearly recover full fine-tuning performance, it seems like there’s enough redundancy with this grafting scheme to extract all the necessary features. Also, to clarify, we graft model components (not directions).
>
> Please let us know if there are further clarifications or improvements that would help the paper, and thanks for your engagement.

---

### Author Response · Authors · 2025-11-30

We offer a comment summarizing the main points of the rebuttal discussion, and we acknowledge the reviewers for their feedback, questions, and comments. We are sorry that we were not able to continue the discussion!

**Synthetic Data & Generalization**: Multiple reviewers raised concerns about the synthetic templated data used in our experiments. While templated data allowed us to control what information models were exposed to during finetuning (things like entity order and syntactic diversity matter a lot), during the discussion period we also ran additional experiments on Wikipedia articles released after the model’s release date, and our results remain consistent.  Grafting on the first entity and the last token nearly recover full finetuning performance, grafting everything else performs the same as the pretrained model; grafting individual pathways is somewhat weaker, but still recovers some performance.

**Choice of Metric**: We chose top-5 accuracy as a simple, intuitive metric for presenting our results in the main body of our paper. Accuracy scales with k for different choices of top-k, but the results are essentially the same. Other choices of metrics also have issues: target token rank is less intuitive and is a skewed distribution, and logit difference is noisy and poorly defined for runs using different models.

**Summary of Contribution**: Reviewers agreed that dynamic weight grafting is a novel methodology, and that the localization of relation knowledge retrieval for the “recall” pathway to the output projection matrix and FFNs is compelling. Our results show that four models with materially different architectures all use similar mechanisms, albeit with some model-specific preferences, to retrieve finetuned factual knowledge.

---

### Meta-Review · Area_Chair_WVcV · 2026-01-21

**Summary:**

This paper introduces a novel method for localizing how newly fine-tuned factual knowledge is retrieved by language models, and apply the technique to demonstrate two complementary mechanisms for retrieving relational information: enchrichment from entity tokens, and recollection at the final token.  Reviewers agreed that the method is novel, the findings are robust across several models, and that the enrichment/recall pathways offer a valuable picture of how fine-tuning affects inference.

**Reviewer Concerns:**

addressed:
- concerns about just using synthetic data were addressed by using Wikipedia data
- clarity of the scope of the claims was given in the rebuttal (single hop relations)

outstanding:
- how differences in the dominance of enrichment vs recall might be explained by architectural differences.
- justification for choice of metric (this seems a matter of taste, though)

**Reviewer Scores:**

reviewers would have seen the concerns about synthetic data were addressed with the Wikipedia experiments

---

### Decision · Program_Chairs · 2026-01-26

Accept (Poster)